# Direct supercritical angle localization microscopy for nanometer 3D superresolution

Anindita Dasgupta[1,2,3,8], Joran Deschamps[1,8], Ulf Matti [1], Uwe Hübner[3], Jan Becker[3], Sebastian Strauss [4,5], Ralf Jungmann [4,5], Rainer Heintzmann[3,6,7] & Jonas Ries [1✉]

3D single molecule localization microscopy (SMLM) is an emerging superresolution method for structural cell biology, as it allows probing precise positions of proteins in cellular structures. In supercritical angle localization microscopy (SALM), z-positions of single fluorophores are extracted from the intensity of supercritical angle fluorescence, which strongly depends on their distance to the coverslip. Here, we realize the full potential of SALM and improve its z-resolution by more than four-fold compared to the state-of-the-art by directly splitting supercritical and undercritical emission, using an ultra-high NA objective, and applying fitting routines to extract precise intensities of single emitters. We demonstrate nanometer isotropic localization precision on DNA origami structures, and on clathrin coated vesicles and microtubules in cells, illustrating the potential of SALM for cell biology.

[1] Cell Biology and Biophysics, European Molecular Biology Laboratory, Heidelberg, Germany. [2] Institute of Applied Optics and Biophysics, Friedrich-Schiller-University, Jena, Germany. [3] Leibniz Institute of Photonic Technology, Jena, Germany. [4] Faculty of Physics and Center for Nanoscience, Ludwig Maximilian University, Munich, Germany. [5] Max Planck Institute of Biochemistry, Martinsried, Germany. [6] Institute of Physical Chemistry, Friedrich-Schiller-University, Jena, Germany. [7] Abbe Center of Photonics, Friedrich-Schiller-University, Jena, Germany. [8] These authors contributed equally: Anindita Dasgupta, Joran Deschamps. ✉email: jonas.ries@embl.de

For many biological questions the 3D organization of proteins is of high interest, therefore SMLM[1–3] has been extended early on to go beyond 2D projections and to measure the 3D coordinates of single fluorescent emitters. This is most commonly achieved either by imaging fluorophores simultaneously in two or more focal planes[4,5] or by introducing aberrations, such as astigmatism[6], which allows extracting the z position from the shape of the point-spread function (PSF). Both approaches are robust and easy to implement, but their resolution is typically 3-fold worse in z and 1.5-fold worse in x and y compared to the lateral resolution of 2D SMLM[7,8]. Interferometric approaches such as iPALM[9] or 4Pi-SMS[10,11] can achieve a superior isotropic resolution, but are complicated to build and to use.

Another conceptually very different approach to determine z coordinates of single fluorophores relies on detecting their supercritical angle fluorescence (SAF, Fig. 1a). SAF is a near-field effect that occurs if a fluorophore is in the vicinity of an interface with a higher refractive index (e.g., the microscope coverslip), and is near-exponentially decaying with the distance from the interface on the scale of 100 nm (Fig. 1b)[12,13]. Due to the blinking and the randomness of activation and deactivation of fluorophores in SMLM, their absolute intensities are not well defined. In order to

reliably extract their z positions, the undercritical angle fluorescence (UAF) has to be collected simultaneously and used for normalization. A major advantage of this approach is that the extracted z coordinates report the absolute distance of the fluorophore from the coverslip.

This idea was implemented in proof-of-concept studies as Supercritical-Angle Localization Microscopy (SALM)[14] or Direct Optical Nanoscopy with Axially Localized Detection (DONALD)[15] in the so-called virtual SAF configuration[16], i.e., by splitting the emission with a 50:50 beam splitter before detecting the total (UAF + SAF) emission in one emission channel and the UAF in a second channel after blocking the SAF with a mask (Fig. 1c). In the following we will refer to this implementation as 'virtual SALM' or vSALM. Although, in theory, vSALM has a higher resolution than astigmatism-based 3D SMLM in the close vicinity of the coverslip (Fig. 1e), experiments on biological samples failed to demonstrate this[14,15,17]. To extend the axial range of vSALM, it has been combined with astigmatism by introducing a cylindrical lens in the UAF channel[18]. Interestingly, SAF has a strong impact on the shape of a defocused PSF. Thus, it can be exploited for precise 3D localization in a simple single-channel setup using an ultrahigh NA objective[19]. However, this approach requires precise knowledge of the z-dependent PSF, which is not easy to calibrate

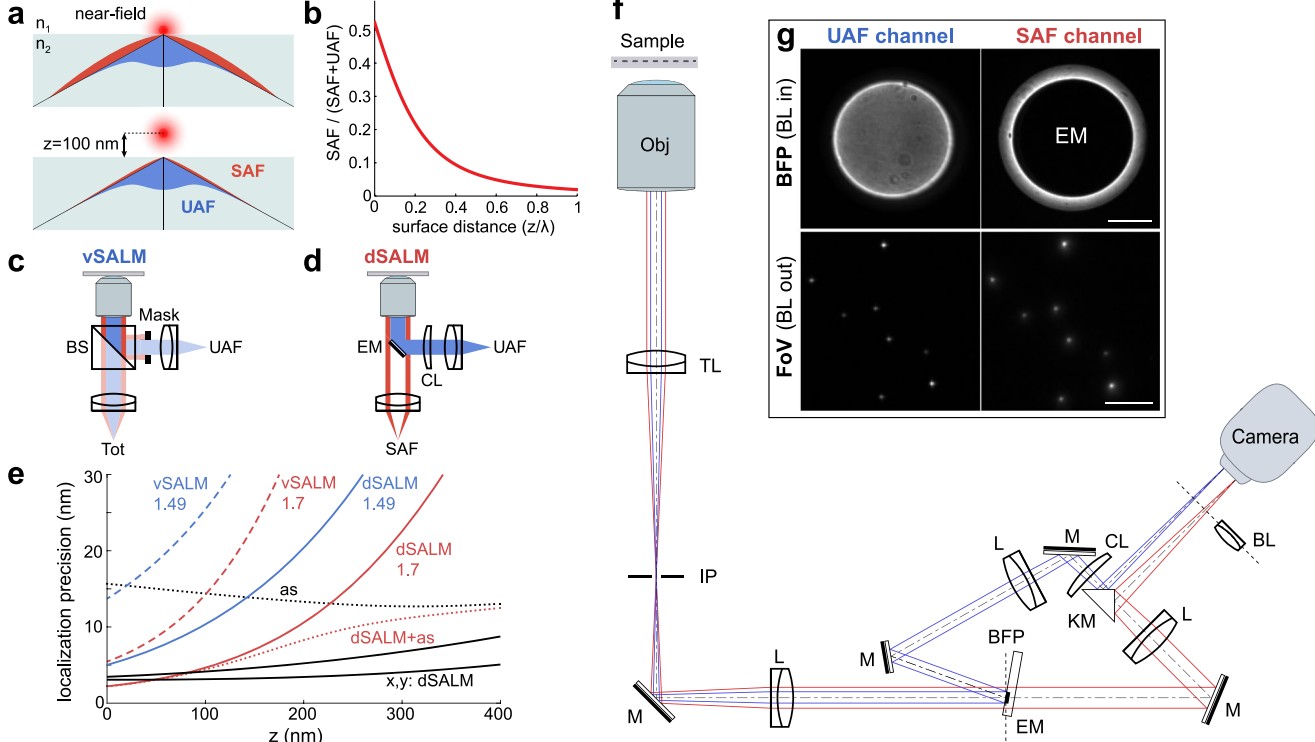

**Fig. 1 Concept of supercritical angle localization microscopy (SALM). a** Supercritical angle fluorescence (SAF) occurs when a fluorophore is close to a water-glass interface. **b** SAF depends strongly on the fluorophore's z position. **c** Virtual SALM (vSALM) splits the emission into two equal parts with a beam splitter (BS) and blocks out SAF with a mask in one channel. The z position of the fluorophore is calculated from the ratio of total fluorescence to undercritical angle fluorescence (UAF). **d** Direct SALM (dSALM) splits SAF and UAF with an elliptical mirror (EM). An additional cylindrical lens (CL) in the UAF channel introduces weak astigmatism (as). **e** Theoretical localization precision in the axial direction calculated for an NA 1.49 objective in the vSALM configuration as used in refs. [14,15] (blue dashed line) and in the dSALM configuration (blue line), and for an NA 1.70 objective for vSALM (dashed red line) and dSALM (red line), astigmatism (black dotted line) and weighted average of dSALM and astigmatism (red dotted line). The solid black line indicates the lateral localization precision of dSALM (NA 1.70) in x and y, respectively. Calculations are based on experimentally derived PSFs (except dSALM 1.49, which is based on a theoretical PSF model) and assume 5000 photons detected in the UAF channel, and a background of 50 photons per pixel in the UAF channel and five photons in the SAF channel. **f** Beam path for the dSALM implementation splitting SAF (red beam) and UAF (blue beam) before forming images of the sample on the camera. BFP back focal plane, BL Bertrand lens, CL cylindrical lens, EM elliptical mirror, IP image plane, KM knife-edge prism mirror, L lens, M mirror, Obj objective, TL tube lens. **g** UAF and SAF channels as seen on the camera with the Bertrand lens (BL) inserted in the beam path (upper panel) or without (lower panel). Scale bars are 2 mm (upper panel) and 5 μm (lower panel). Source Data are available as a Source Data file for Fig. 1e.

in presence of aberrations. Additionally, the required defocusing leads to a loss in lateral resolution. Related approaches used the enhanced near-field emission at metal[20,21] or dielectric[22] interfaces to precisely localize biological structures axially.

Direct splitting of SAF and UAF in SALM (direct SALM or dSALM, Fig. 1d) promises a several-fold improved z resolution and useable depth of field above the coverslip compared to vSALM (Fig. 1e, Supplementary Fig. 1). The SAF intensity can be quantified much more precisely if detected by itself than as part of the total intensity (SAF + UAF, as in vSALM), in particular when SAF is weak at larger z positions (see Methods, theory). The reason lies in the fact that the main source of noise is shot noise, therefore the relative error in determining photon numbers $N$ scales with $\delta N/N \sim N^{-1/2}$. Additionally, and in contrast to vSALM, where splitting of the fluorescence halves the UAF signal, the entire UAF signal is used for normalization in dSALM, further improving precision. Finally, in vSALM half of the SAF signal is discarded. However, direct splitting of SAF and UAF had not been realized in SMLM because blocking the UAF in the SAF channel results in strong diffraction patterns dominating the SAF PSF[23] (Supplementary Fig. 2), leading to an increased PSF size and preventing a reliable measurement of single-molecule intensities.

Here, we overcome these challenges and realize the full potential of SALM by combining (a) direct measurement of SAF with greatly increased signal to noise ratio by splitting it from UAF with a custom elliptical mirror, (b) use of an ultrahigh NA objective to increase the SAF signal and to decrease the effect of diffraction on the SAF PSF, and (c) data analysis approaches that allow for precise determination of UAF and SAF intensities even in presence of a complex PSF. We demonstrate on 3D DNA origami structures that dSALM can fulfill its potential in terms of 3D resolution and show on biological samples that a combination with astigmatism leads to a remarkable z resolution over an extended axial range.

## Results
**dSALM principle**. In the following, we give a short overview of the dSALM implementation, details can be found in the Methods section. To measure SAF and UAF of single fluorophores, we split the fluorescence in a plane conjugated to the objective back focal plane (BFP) with a custom-made elliptical mirror (Supplementary Fig. 4) and image both emission paths side-by-side on two halves of a camera (Fig. 1f). In the undercritical path, we insert a cylindrical lens to introduce weak astigmatism[18]. By fitting single molecules in the UAF channel with an experimentally derived PSF model[17], we obtain their precise x and y coordinates and intensity, as well as less precise z coordinates (referred to as $z_{as}$). Using these coordinates and a predetermined transformation between the two channels, we fit an experimental SAF PSF model to the single-molecule images in the SAF channel to extract their precise SAF intensities. From the ratio of SAF and UAF intensities we then calculate precise and absolute z positions, referred to as $z_{dSALM}$, using the theoretical dependence of the SAF signal on z (Eq. 4). For samples with an extended z range, we calculate $z_{av}$ as a weighted average of $z_{dSALM}$ and $z_{as}$[18] (see Eq. 13, Methods, Data analysis section).

**Experimental localization precision**. We validated our imaging and data analysis workflow on 50-nm fluorescent beads, immobilized on the coverslip (Fig. 2). By imaging the same bead multiple times, we estimated the experimental localization precision as the standard deviation of the coordinates and compared it to the calculated localization precision (see Eq. 7, Methods). The calculated localization precision faithfully represented the

experimental one for a wide range of excitation intensities (Fig. 2c).

**Validation using DNA origamis**. To quantify the resolution of dSALM, we used DNA-PAINT[24] to image DNA origami nanorulers that consisted of two rings of DNA binding sites separated by a distance of 30-nm (Fig. 3a–c). The two rings were easily resolved in side-view reconstructions, with an average number of photons per localization of 21,300, and a fit of z-profiles resulted in a standard deviation of z positions of $3.2 \pm 1.4$ nm and $5.8 \pm 1.1$ nm for the lower and upper ring, respectively (Fig. 3d). Due to possible deformations of the DNA structures and residual tilt, these are worst-case estimates for the experimental localization precision, the best-case estimate is the localization precision calculated from the Cramér-Rao Lower Bound (CRLB, Methods), which peaks at 2.7 nm (Fig. 3e). In astigmatism-based SMLM, aberrations lead to a mismatch between model and data and consequently to systematic localization errors that depend on the distance of the fluorophore to the coverslip[25,26]. On the other hand, the intensity ratio between SAF and UAF is an absolute measure for the distance of the fluorophore from the coverslip and should be less sensitive to these aberrations. To test this, we imaged DNA origami rulers that consisted of two rings of DNA binding sites with a distance of 80-nm (Fig. 3f, g). Their immobilization on the coverslip resulted in a wide distribution of angles of the rulers with respect to the image plane. Any systematic localization errors would introduce a correlation between the angle with respect to the z-axis and the measured distance, something apparent in astigmatic 3D SMLM[25] (Fig. 3h). In contrast, the measured distance based on dSALM showed a much smaller dependence on the angle, validating that dSALM allows for absolute z distance measurements with greatly reduced systematic errors. Residual variations in the measured length could be due to the inherent size distribution of the DNA origami rulers and bending[25], as well as residual flexibility in their attachment to the coverslip.

**Imaging cellular structures**. To demonstrate the usefulness of our approach for biological research, we used speed-optimized DNA-PAINT[27] to image two cellular structures, clathrin coated pits that form spherical assemblies with a diameter of around 150 nm (Fig. 4a, b, Supplementary Fig. 5), and microtubules, where the antibody labels form cylindrical arrangements with a diameter of ~50 nm around the filaments[28] (Fig. 4c, d). These geometries are well visible in the side-view reconstructions (Fig. 4b, d). To extend the z range beyond the range accessible with dSALM, we based these reconstructions on $z_{av}$. Compared to using $z_{dSALM}$ alone, the z resolution away from the coverslip is improved (Fig. 4b1). Note that in this range, systematic position errors caused by depth-dependent aberrations reappear (compare Fig. 3h).

## Discussion
The main limitation of dSALM is the limited axial range of a few hundred nanometers above the coverslip, making it a robust technique to study processes in or close to the plasma membrane of cells. Reaching a similar resolution deeper inside cells requires much more complex and less robust microscopes such as 4Pi-SMLM[9,11] or 3D MINFLUX[29], which are out of reach for most researchers due to their high cost.

Since SAF depends on the orientation of the transition dipole moment of the fluorophore, dSALM should only be used with standard labeling approaches using fluorescent proteins, antibodies or self-labeling enzymes that allow for free rotation of the fluorophores. This is the case for DNA-PAINT, where the

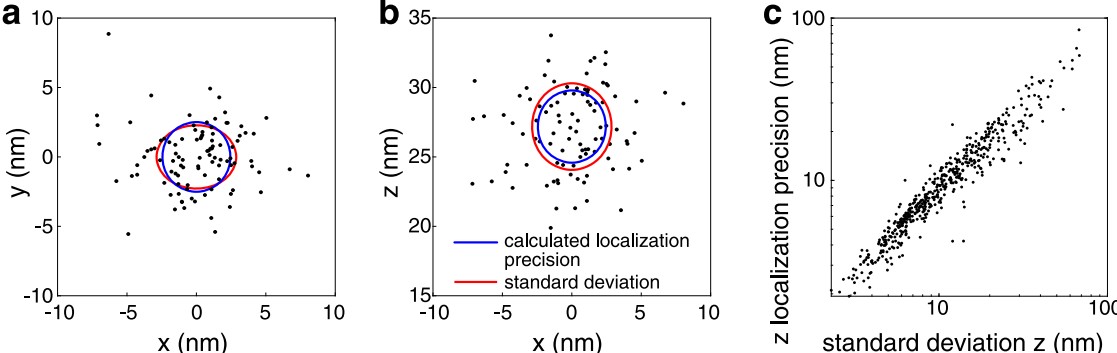

**Fig. 2 Validation using fluorescent beads.** 50-nm red beads were immobilized on the coverslip and imaged 100 times to calculate the experimental localization precision as the standard deviation of the coordinates. **a** x–y scatter plot of the fitted localizations of a single bead, **b** x–z scatter plot. z coordinates are based on dSALM alone. The red line denotes the experimental localization precision calculated as the standard deviation of the coordinates. The blue line denotes the average calculated localization precision based on the Cramér-Rao-Lower Bound (Eq. 7, Methods). **c** A comparison of calculated and experimental localization precision in z for 525 beads in two independent experiments, measured with different excitation laser powers spanning two orders of magnitude, leading to a brightness of the beads between 150 and 15,000 photons. Source Data are available as a Source Data file for Fig. 2.

fluorophores are coupled to the imaging strand by a flexible linker and the docking strand contains unpaired nucleotides. The high refractive index immersion oil, required by the ultrahigh NA objective, quickly degrades with blue or ultraviolet excitation, currently limiting the use of dSALM to SMLM with self-blinking dyes. Thus, the development of improved immersion oil is highly desirable to extend dSALM to photoactivatable fluorophores commonly used in SMLM. Finally, the circular field of view is limited to a diameter of 20 μm due to strong field-dependent aberrations of the objective. A new generation of ultrahigh NA objectives could increase the field of view and, thus, throughput. Finally, dSALM makes use of a custom-made elliptical mirror whose availability is currently limited.

To summarize, we developed dSALM, a 3D SMLM method that reaches a localization precision in z better than 4 nm near the coverslip, while retaining the high lateral resolution of 2D SMLM and providing absolute z distances from the coverslip. At this high optical resolution, the size of the label and the density of labeling are likely to become the limiting factor. Compared to vSALM, a previous implementation of the method, the resolution and useable range above the coverslip are improved by more than 4-fold. Thus, we expect dSALM to find widespread use in cell and structural biology and to enable new discoveries that are currently not accessible with standard SMLM.

## Methods
### Theory: extracting single fluorophore z positions from supercritical angle fluorescence.
A fluorophore close to a water-glass interface can couple its emission directly into the glass in an effect called surface-generated fluorescence or super-critical angle fluorescence[30,31]. This fluorescence strongly depends on the distance of the fluorophore from the interface. For freely rotating molecules, its intensity in the SAF channel $I_S(z)$ can be calculated numerically[30,32]:

$$I_S(z) = \int_0^{\nu_{NA}} \frac{2\left(n_1^2 + n_2^2\right)\nu\lambdabar\sqrt{n_2^2 - n_1^2 - \nu^2\lambdabar^2}\left(n_1^2 + \nu^2\lambdabar^2\right)}{3\left(n_2^2 - n_1^2\right)\left(n_1^4 + \left(n_1^2 + n_2^2\right)\nu^2\lambdabar^2\right)} e^{-2\nu z}d\nu \quad (1)$$

Here $\lambda$ is the wavelength of the emitted light, $\lambdabar = \lambda/2\pi$, $n_1$ and $n_2$ are the respective refractive indices of the buffer and the glass coverslip, $\nu_{NA} = \lambdabar^{-1}\left(NA^2 - n_1^2\right)^{-1/2}$, and the numerical aperture of the objective is defined as $NA = n_2 \sin\Theta_{NA}$.

The intensity in the UAF channel $I_U$ is

$$I_U = \int_0^{w_0} \frac{2Q\lambdabar w}{3}\left(\frac{1}{(w\lambdabar + Q)^2} + \frac{n_1^2 n_2^2}{(n_1^2 Q + n_2^2 w\lambdabar)^2}\right)dw \quad (2)$$

with $Q = \left(n_2^2 - n_1^2 + w^2\lambdabar^2\right)^{1/2}$ and $w_0 = n_1/\lambdabar$.

In dSALM, we experimentally detect the number of photons emitted by a single fluorophore in the SAF channel as $N_S$ and in the UAF channel as $N_U$. To extract the

z position of the fluorophore, we first calculate the theoretical intensity ratio from Eqs. (1) and (2):

$$f(z) = \frac{I_S(z)}{I_U} \quad (3)$$

We then numerically invert this relationship and approximate $f^{-1} = z(f)$ with a cubic spline. Then we can directly transform $N_S$ and $N_U$ to the fluorophore's z position:

$$z_{dSALM} = f^{-1}\left(\frac{N_S(z)}{N_U}\right) \quad (4)$$

**Calculating the Cramer-Rao Lower Bound (CRLB).** The CRLB is a best-case estimator of the precision of the fitting parameters. We use it to calculate the theoretical precision of dSALM and vSALM (Fig. 1e), and to assign experimental localization precisions to every single molecule. The CRLB can be calculated from the inverse of the Fisher Information matrix FI$u,\nu$[19,33]:

$$CRLB_u = \left[FI^{-1}\right]_{uu} \quad (5)$$

with

$$FI_{u,\nu} = \sum_{k=1}^{K} \frac{1}{\mu_k}\frac{\partial\mu_k}{\partial\theta_u}\frac{\partial\mu_k}{\partial\theta_\nu}. \quad (6)$$

$\mu_k(\theta_u)$ is the model describing the intensity in each pixel k. In our case, it is an experimentally derived spline-interpolated PSF model. $\theta_u = \{x, y, z_{as}, N, b\}$ are the fitting parameters that include the position of the fluorophore $x, y, z_{as}$, the number of photons $N$ and the background per pixel $b$.

For dSALM, we estimate the lateral localization precisions $\delta x = \sqrt{CRLB_x}$ and $\delta y = \sqrt{CRLB_y}$ and the axial PSF-based localization precision $\delta z_{as} = \sqrt{CRLB_{z,as}}$ from the CRLB of the UAF channel only. The photometry-based axial localization precision in dSALM we calculate from the precision $\delta N_S = \sqrt{CRLB_{N,S}}$ and $\delta N_U = \sqrt{CRLB_{N,U}}$ of the number of photons detected in each channel using Eq. 5. To this end, we start with Eq. 3, use the definition $f(z) = N_S/N_U$ and apply Gaussian error propagation[19]:

$$\delta z_{dSALM} = \left(\frac{\partial f(z)}{\partial z}\right)^{-1}\delta f(z), \quad (7)$$

$$\delta f(z)^2 = \left(\frac{\partial f(z)}{\partial N_S}\delta N_S\right)^2 + \left(\frac{\partial f(z)}{\partial N_U}\delta N_U\right)^2 \\ = \left(\frac{1}{N_U}\delta N_S\right)^2 + \left(-\frac{N_S}{N_U^2}\delta N_U\right)^2. \quad (8)$$

The combined axial localization precision from photometry $\delta z_{dSALM}$ and PSF shape $\delta z_{as}$ we calculate as the weighted average of each localization precision (compare with Eq. 13 in the Data analysis section):

$$\delta z_{av} = \left(\delta z_{dSALM}^{-2} + \delta z_{as}^{-2}\right)^{-1/2}. \quad (9)$$

For vSALM, we estimated the lateral localization precision as the weighted average of the localization precisions of the UAF and total fluorescence

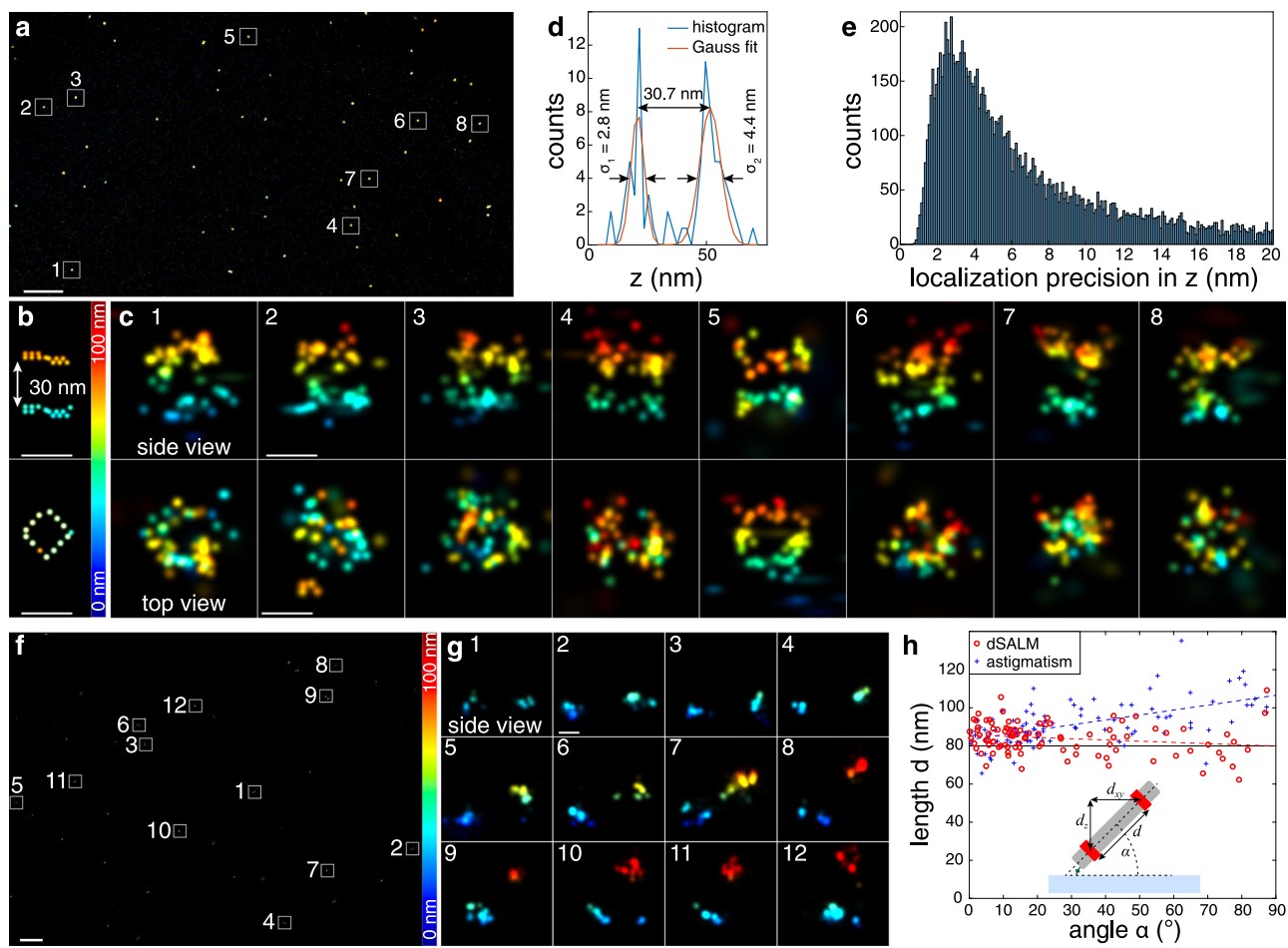

**Fig. 3 Validation with DNA origamis. a** dSALM performed on 30-nm DNA origami rulers to test precision, overview image. **b** Schematic: each dot represents the position of a fluorophore binding site on the origami, as per design. **c** Side-view and top view reconstructions of the rulers corresponding to the marked regions in **a**. Localizations are color-coded according to their z positions ($z_{SALM}$). **d** Axial profile through the structure c7 with Gaussian fits resulting in a standard deviation of 2.8 nm for the lower and 4.4 nm of the upper ring. The average standard deviations of all structures c1–7 are 3.2 ± 1.4 nm and 5.8 ± 1.1 nm for the lower and upper ring, respectively, and their average distance is 27.6 ± 1.6 nm. The average number of photons per localization is 21,300. **e** The histogram of axial dSALM localization precisions ($\delta z_{dSALM}$, Eq. 7, Methods) peaks at 2.7 nm. **f** dSALM performed on 80-nm DNA origami rulers to test accuracy, overview image. **g** Side-view reconstructions corresponding to marked regions in **f**. **h** Measured length of the 80-nm DNA origami rulers in dependence on the angle shows strong depth-dependent deformations for astigmatism (blue, Pearson correlation coefficient 0.58), and much smaller deformations for dSALM (red, Pearson correlation coefficient −0.19). We measured an average length of 91.6 ± 11.5 nm with astigmatism, and 83.8 ± 8.3 nm with dSALM (N = 94, from three independent replicates). Inset: sketch of the 80-nm DNA origami ruler. Scale bars: 1 µm (**a**, **f**), 30 nm (**b**, **c**, **g**). Numbers denote mean ± standard deviation. Both **a** and **f**, and their respective subregions in **c** and **g**, are representative images from two (**a**) and three (**f**) independent experiments. Source Data are available as a Source Data file for Fig. 3e,h.

channel, e.g.,:

$$\delta x = \left( \delta x_U^{-2} + \delta x_{Tot}^{-2} \right)^{-1/2}. \tag{10}$$

The axial localization precision based on photometry we calculate analogous to the dSALM example but with $N_S(z)$ replaced by $N_{Tot}(z)$ and $f_v(z) = I_{Tot}(z)/I_U = f(z) + 1$:

$$\delta f_v(z)^2 = \left( \frac{1}{N_U} \delta N_{Tot} \right)^2 + \left( -\frac{N_{Tot}}{N_U^2} \delta N_U \right)^2. \tag{11}$$

If we approximate $\delta N$ with $\sqrt{N}$, we can directly see where the improvement of dSALM vs vSALM comes from:

$$\delta f(z)^2 = \frac{N_S}{N_U^2} + \frac{N_S^2}{N_U^3}$$
$$\delta f_v(z)^2 = \frac{N_{Tot}}{N_U^2} + \frac{N_{Tot}^2}{N_U^3} \tag{12}$$

First, in vSALM the beam splitting leads to a decrease in $N_U$ by a factor of 2 compared to dSALM. Second, especially for large z, $N_{Tot} \gg N_S$. Both increase $\delta f_v$ compared to $\delta f$.

For calculating the CRLB in Fig. 1e, we assumed that we collect $N_c = 5000$ photons from a fluorophore far away from the coverslip. For fluorophores with a low quantum yield (as is the case for our experimental data with a quantum yield on the order of 30%) SAF competes not only with UAF, but mostly also with the

nonradiative decay. Thus, we made the approximation $N_U = N_c$ and $N_S(z) = f(z)N_c$. For vSALM, after beam splitting $N_U = N_c/2$ and $N_{Tot}(z) = (N_U + N_S(z))/2$. For fluorophores with high quantum yield, SAF competes with UAF and $N_U + N_S = N_c$. Then $N_U(z) = N_c/(1 + f(z))$ and $N_S(z) = N_c f(z)/(1 + f(z))$. The curves in Fig. 1e, although different in details, remain qualitatively the same.

**Microscope.** The detection beam path is shown in Fig. 1f. A laser combiner (iChromeMLE, Toptica) delivers the lasers (405, 488, 561, and 640 nm) to the microscope via a single-mode optical fiber. The laser beam is subsequently collimated by an achromatic lens (75 mm, Thorlabs) and circularly polarized by a quarter waveplate (Thorlabs). Another achromatic lens (dubbed illumination lens, 175 mm, Thorlabs), in 4 f configuration with the objective, focuses the laser on the objective BFP. The fiber output is mounted on a linear stage (SmarAct) in order to switch the illumination mode between total internal reflection (TIR), highly inclined and laminated optical sheet (HiLo) or epi-illumination. Alternatively, a custom laser engine can be used to obtain homogeneous illumination by agitating a multimode optical fiber[34]. After the illumination lens, the lasers are reflected on a 4× dichroic mirror (F73-410, AHF) before the objective (NA 1.7 APON 100XHOTIRF, Olympus). The objective is housed in a z-stage (P-726, PI), while the sample position is controlled by a 2D stage (SmarAct). Fluorescence is collected by the objective and propagates through the 4× dichroic mirror. An intermediate image of the sample is formed by the tube lens (U-TLU, Olympus) outside of the

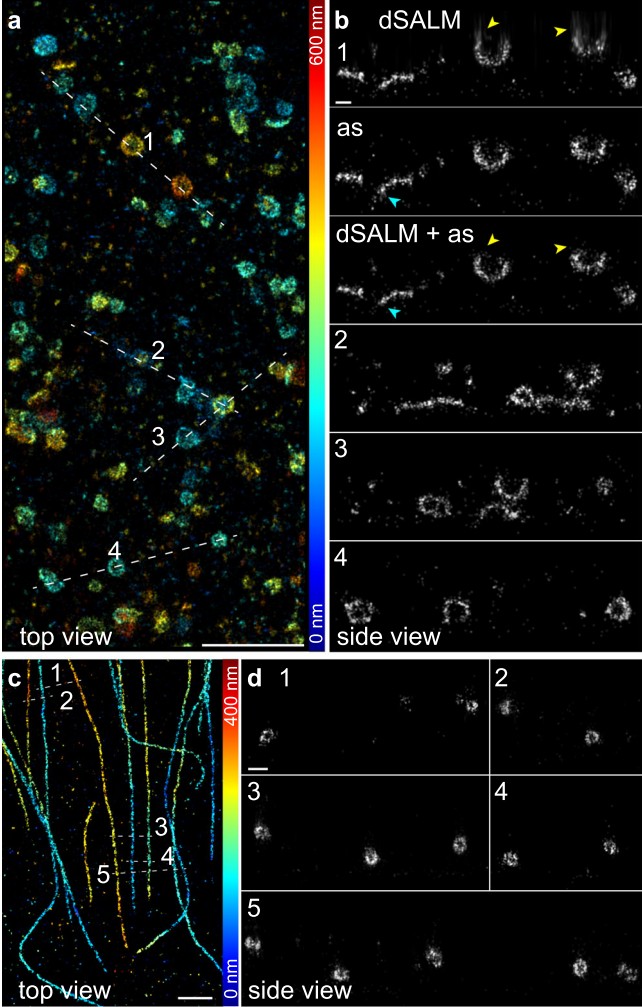

**Fig. 4 Application to biological structures. a** Clathrin coated pits, labeled with anti-clathrin heavy and light chain primary and speed-optimized DNA-PAINT secondary antibodies. Localizations are color-coded according to their z positions. **b** Side-view reconstructions of regions as denoted in **a** (width of the line ROIs 50 nm). **b1** shows a comparison of a dSALM reconstruction ($z_{dSALM}$), a reconstruction based on astigmatism ($z_{as}$), and a combined dSALM + astigmatism reconstruction ($z_{av}$). Yellow arrows indicate regions where the combined image shows an improved resolution compared to dSALM alone, cyan arrows indicate regions where the combined image shows a higher resolution than astigmatism alone. **b2–4** are based on $z_{av}$. See Supplementary Fig. 5 for additional reconstructions and localization precision histograms. **c** Microtubules, labeled with anti alpha-tubulin primary and speed-optimized DNA-PAINT secondary antibodies. **d** Side-view reconstructions of regions as denoted in **c** (width of the line ROIs 200 nm). Scale bars: 1 μm (**a**, **c**), 100 nm (**b**, **d**). Both **a** and **c** are representative images from two independent experiments each.

microscope body. Another achromatic lens (200 mm, Thorlabs) in 4 f configuration gives access to a plane conjugated with the objective BFP. In this plane, we inserted the elliptical mirror housed in a 2D translation mount (Thorlabs) at an angle of 10°, giving rise to the SAF and UAF channels. Finally, two achromatic lenses (250 mm, Thorlabs), in 4 f configuration with the previous one, image the two channels side-by-side on two halves of an EMCCD camera (iXon 897, Andor). In the reflected channel (UAF), a cylindrical lens with 2 m focal length (SCX-50.8-1000.0-UV, CVI Laser Optics) is inserted to introduce weak astigmatism. Active axial drift correction is implemented by using a 785-nm laser (iBeamSmart, Toptica) coupled into the objective and reflected in TIR at the coverslip. The near infrared laser is then detected by a quadrant photo-diode (QP50-6 TO, First Sensor) and an analog signal from the sensor's amplifier (LC-301DQD-PV, Laser Components) is directly fed to the objective z-stage controller. The microscope is entirely controlled by Micro-Manager[35] using htSMLM, a custom EMU plugin[36].

**Elliptical mirror**. The mirror was produced at the Micro- and Nanotechnology center at the Leibniz institute, Jena, Germany. We coated a layer of 65-nm protective $Al_2O_3$ onto 200-nm silver coating on a 5-mm thick AR-coated BK-7 glass (WG11050-A, Thorlabs) substrate with λ/10 nominal surface flatness. The layer structure was optimized aiming for a good compromise between predicted reflectivity (94.5% at 660 nm), material absorption and surface flatness. The optimization was performed with the help of a home-written computer program (Supplementary Software). The elliptical mirror used in this manuscript has a diameter of 5.50 × 5.58 mm, designed for a circular footprint when reflecting under an incident angle of 10°. The elliptical mirror was regularly cleaned under a microscope using an eyelash manipulator (commonly used for ultramicrotomy) to remove dust without damaging the surface.

The elliptical mirror is aligned by inserting the Bertrand lens in front of the camera (see Fig. 1f) and centering the mask in the SAF channel in the BFP image (see Fig. 1g). As the mask is slightly larger than the critical angle, even minor misalignments or changes in the refractive index (e.g., due to the presence of the cell) do not lead to undercritical light entering the SAF path.

**Sample seeding on coverslip**. The ultrahigh NA objective requires index-matched 20-mm round high refractive index coverslips (HIGHINDEX-CG, N4247800, Olympus). Since such coverslips are expensive, they were systematically cleaned and reused multiple times. To do so, coverslips were first cleaned gently with a tissue paper (Kimwipes, Kimtech) soaked in a 70% EtOH solution. Next, they were kept overnight in 70% EtOH and finally air-dried for a few hours prior to the seeding of cells onto the coverslips. Then, cells (human bone osteosarcoma, U2OS) were seeded onto the coverslips 24–48 h before fixation in such a density that they reach a confluency of 50–70% before further processing. Cells were grown in an incubation chamber providing 37 °C and 5% $CO_2$ in growth medium (DMEM, catalog no. 11880-02, Gibco) containing 1× MEM NEAA (catalog no. 11140-035, Gibco), 1× GlutaMAX (catalog no. 35050-038, Gibco), and 10% (v/v) fetal bovine serum (catalog no. 10270-106, Gibco). Finally, shortly before fixation (see specific sample type description), coverslips were rinsed twice with warm PBS.

**Imaging buffer for speed-optimized DNA-PAINT imaging**. The following imaging buffer was used for imaging of microtubules and clathrin coated pits[27]. It is comprised of 1× Trolox, 1× PCA, 1× PCD, 75 mM MgCl$_2$, 5 mM Tris pH 8, 1 mM EDTA, 0.05% Tween 20. Herein, 100× Trolox stock comprised of 100 mg Trolox in 430 μL methanol, 345 μL 1 M NaOH, 3.2 ml $H_2O$. 40× PCA stock comprised of 154 mg PCA in 10 ml $H_2O$ (adjusted to pH9.0 with 1 M NaOH). 100× PCD stock comprised of 9.3 mg PCD in 13.3 ml (100 mM Tris pH 8, 50 mM KCl, 1 mM EDTA, 50% Glycerol).

**100-nm bead sample**. Fluorescent beads were deposited on the coverslip by first preparing a solution with 360 μL $H_2O$ and 1 μL of fluorescent beads (0.1 μm TetraSpeck, T7279, Thermo Fisher Scientific). The solution was vortexed thoroughly. Meanwhile, 40 μL of 1 M MgCl$_2$ were pipetted onto the coverslip. Then, the beads solution was added to the coverslip and mixed. After 10 min the buffer was removed and 400 μL $H_2O$ was added to the coverslip.

**50-nm bead sample**. Fluorescent beads were diluted 1000 times in $H_2O$ from the original vial (0.05 μm FluoSpheres, Lot 2129000, Thermo Fisher Scientific). From this dilution 0.5 μL of beads were added to 450 μL $H_2O$. The solution was vortexed thoroughly. Meanwhile, 50 μL of 1 M MgCl$_2$ were pipetted onto the coverslip. Then, the beads solution was added to the coverslip and mixed. After 10 min the buffer was removed and 450 μL $H_2O$ and 50 μL of 1 M MgCl$_2$ was added to the coverslip.

**30-nm DNA origami nanoruler sample**. In order to image the 30-nm 3D nanorulers (GATTA-PAINT 3D 30 R ultimate line, GATTAquant), the high NA coverslips were washed three times with 500 μL of PBS. Then the coverslips were incubated with 200 μL of BSA-biotin solution (1 mg/ml in PBS) for 5 min. Upon removing the BSA-biotin solution, the coverslips were washed three times with 500 μL of PBS. Care was taken to ensure that the surface of the coverslips did not get scratched with the pipette tip. The coverslips were incubated with 200 μL of neutravidin solution (1 mg/ml in PBS) for 5 min. Upon removing the neutravidin solution the coverslips were washed three times with 500 μL of 1× PBS supplemented with immobilization buffer (1× IB: 10 mM MgCl$_2$ + 500 mM NaCl + 50 mM Tris pH 8). 7.5 μL of the 30 nm DNA origami solution was diluted with 200 μL 1× IB. The coverslips with the origami solution were incubated for 5 min. Eventually the coverslips were washed three times with 500 μL of 1× IB. The coverslips were then mounted into custom sample holders with 800 pM of imaging DNA strands coupled to Atto655 in imaging buffer comprised of 50 mM Tris pH 8, 10 mM MgCl$_2$, 500 mM NaCl.

**80-nm DNA origami sample**. In the case of the 80-nm nanorulers (GATTA-PAINT 3D HiRes 80 R, GATTAquant), the coverslips were treated following the same steps as described above for the 30-nm DNA Origami preparation. Ten microliter of the 80 nm DNA origami solution was diluted with 5 μL 1× IB. The coverslips with the origami solution were incubated for 5 min. Eventually the

coverslips were washed three times with 500 µL of 1× IB. The coverslips were then mounted into custom sample holders with 500 pM of imaging DNA strands coupled to Atto655 in imaging buffer comprised of 50 mM Tris pH 8, 10 mM MgCl$_2$, 500 mM NaCl.

**Indirect immunostaining of clathrin coated pits.** Cells on coverslips were fixed at 37 °C with 3% formaldehyde (FA) in cytoskeleton buffer (10 mM MES, pH 6.2, 150 mM EGTA, 5 mM D-Glucose, 5 mM MgCl$_2$) for 20 mins on a shaker. Then, the samples were rinsed once with 2 ml of freshly prepared 0.1% NaBH$_4$ in PBS and then quenched in 0.1% NaBH$_4$ in PBS for 7 min. After this, the samples were washed three times in PBS for 5 min each. In order to permeabilize the membranes, the samples were incubated in 100 µL of 0.01% Digitonin (Sigma D141) in PBS for 15 min. This was followed by two times washing in PBS for 5 min each. To block unspecific binding, the samples were placed in 100 µL ImageIT (I36933, Thermo Fisher Scientific) for 30 min and, following that, 30 min in 100 µL 2% BSA in PBS. For primary antibody labelling, anti-clathrin heavy chain rabbit antibody (ab21679, Abcam) (1:300 final dilution) and anti-clathrin light chain rabbit antibody (sc28276, Santa Cruz Biotechnology) (1:100 final dilution) were used in 100 µL 50% v/v ImageIT in PBS containing 1% (w/v) BSA overnight at 4 °C. Subsequently, binding of polyclonal donkey anti-rabbit secondary antibodies (catalog no. 711-005-152, Jackson ImmunoResearch) coupled to a DNA-PAINT docking site (TCCTCCTC) was achieved by placing the samples upside down onto a 1:100 dilution of the antibodies in PBS containing 2% (w/v) BSA overnight at 4 °C. The sample was imaged with in speed-optimized DNA-PAINT imaging buffer containing 17 pM of imager strands with the sequence AGGAGGA-ATTO643 (Eurofins Genomics).

**Indirect immunostaining of microtubules.** Coverslips containing U2OS were prefixed with 0.3% (w/v) glutaraldehyde (GA) in cytoskeleton buffer (CB: 10 mM MES pH 6.1, 150 mM NaCl, 5 mM EGTA, 5 mM D-glucose, 5 mM MgCl$_2$) containing 0.25% (v/v) Triton X-100 for 60 s before samples were incubated in 2% (w/v) GA in CB for 10 min. After briefly washing the samples with PBS, auto-fluorescence from GA fixation was quenched by incubation with freshly prepared PBS containing 0.1% (w/v) NaBH$_4$ for 10 min. The sample was subsequently washed in PBS until no more bubbles were observed, usually 5 × 5 min.

Primary antibody staining was carried out by placing coverslips upside down onto a drop of primary antibody staining mix (mouse-anti-beta-tubulin, catalog no. T5293, Sigma–Aldrich, diluted 1:300 in PBS containing 2% (w/v) BSA) for 2–3 h at RT. Weakly and unbound primary antibodies were washed off three times in PBS for 5 min each. Similarly, polyclonal donkey anti-mouse secondary antibodies (catalog no. 715-005-151, Jackson ImmunoResearch) coupled to a DNA-PAINT docking site (TCCTCCTC) was achieved by placing the samples upside down onto a 1:100 dilution of the antibodies in PBS containing 2% (w/v) BSA overnight at 4 °C. After washing thrice in PBS for 5 min each, post-fixation was carried out in 2.4% (v/v) formaldehyde in PBS for 30 min. Samples were quenched with 100 mM NH$_4$Cl for 10 min, washed twice for 5 min in PBS and finally placed into a custom-made sample holder. The sample was imaged in speed-optimized DNA-PAINT imaging buffer containing 10 pM of imager with the sequence AGGAGGA-ATTO643[27].

**Antibody-DNA conjugation.** Antibodies were conjugated to DNA-PAINT docking sites via DBCO-sulfo-NHS ester chemistry[37]. In brief, antibodies were reacted with 20-fold excess of a bifunctional DBCO-sulfo-NHS ester (Jena Biosciences, cat: CLK-A124-10). Unreacted linker was removed using Zeba Spin Desalting columns (0.5 ml, 40k MWCO, Thermo Fisher Scientific, cat: 89882). Azide-DNA (C3-azide-TCCTCCTC, Metabion) was added to the DBCO-antibodies with a 10-fold molar excess and reacted overnight at 4 °C. Afterwards, buffer was exchanged to PBS using Amicon centrifugal filters (100k MWCO).

**Determining the region of low optical aberrations.** The ultrahigh NA objective from Olympus displays strong field-dependent aberrations. Only a small part at the center of the objective allowed for efficient PSF averaging and precise estimation of intensities. We estimated the position of this region by taking z-stacks on 10 different fields of view without EM gain, from −1.5 µm to +1.5 µm with 50 nm step size. We then fitted the z-stacks using an experimentally derived PSF model and calculated the displacement standard deviations in x and y for each bead. The analysis showed a strong dependency of the displacement standard deviation on the beads distance to the center of the objective. Based on these results, we limited all our experiments to a circle of radius 10 µm around the objective center.

**Preparation of the immersion oil.** Before each series of experiments, 30 µL of high refractive index oil (Series M 1.780, Cargille) was centrifuged at 8610 × g for 2 min, causing oil crystals to sediment. Only 10 µL of the supernatant is used at a time on the objective. When imaging nanorulers and biological samples, the oil was exchanged for every new region of interest.

**Acquiring image stacks with 100-nm beads.** About 15 fields of view were manually chosen, concentrating on regions with few non-overlapping beads. The stacks were recorded at 30 ms exposure time, without EM gain from −1.2 µm to +1.2 µm around the focus in steps of 30 nm. A mean excitation intensity of 0.9 kW/cm$^2$ was used.

**Imaging 50-nm beads.** About 60 fields of view were manually chosen, concentrating on regions with several non-overlapping 50-nm beads. 100 image frames were recorded at 100 ms exposure time using different excitation laser powers between 0.1 and 11 kW/cm$^2$.

**Imaging DNA-PAINT nanorulers.** The 80-nm nanorulers were imaged with an EM gain of 100 and an exposure time of 200 ms. Imaging was performed under HiLo illumination for ~13,000 frames (640 nm laser) at a power density of 12 kW/cm$^2$. The 30-nm nanorulers were recorded with an EM gain of 100 and 500 ms exposure time. Imaging was performed under total internal reflection illumination for 12,000 frames using the commercial laser combiner (640 nm laser) at a power density of 10 kW/cm$^2$.

**Imaging biological samples.** Images of Clathrin-mediated endocytosis were acquired with 100 ms exposure time and an EM gain of 100. We typically imaged the samples for 100,000 frames in HiLo (640 nm laser form the commercial combiner). The illumination led to an estimated power density of 6.5 kW/cm$^2$. Microtubules images were acquired using the same settings, albeit with a slightly lower power density (5.6 kW/cm$^2$).

**Data analysis.** All data analysis was performed with a custom MATLAB software called Superresolution Microscopy Analysis Platform[38] (SMAP, https://github.com/jries/SMAP).

**Calibration of the experimental PSF model.** In dSALM, the z position of a single fluorophore is extracted from its relative intensities in the supercritical and undercritical channel. This requires the precise measurement of these intensities in the camera images, which is challenging because diffraction causes the PSF to be very different from one channel to the other, and depends on z. Here, we overcome this challenge by fitting the fluorophore images with experimentally derived PSF models. To this end we use stacks of beads immobilized on a coverslip. We generated a spline-interpolated PSF model for each channel after registering and averaging about 40 beads[17]. In particular, we only considered beads within a circle of 10 µm radius around the center of the objective field of view, a region where the objective PSF does not display strong field-dependent aberrations. The resulting PSF calibration is used for analysis of all data.

**Quantification of SAF and UAF intensities.** For the nanorulers and the biological samples, the localization microscopy raw data were fitted with the experimental PSF models for the UAF and SAF channel, using maximum likelihood estimation[17]. From the fitted positions of fluorophores detected in both channels, we generated a projective transformation (that describes translation, scaling, rotation, shearing and tilting and is represented by a 3 × 3 matrix with arbitrary elements) from the SAF to the UAF channel. Based on the 3D coordinates of every fluorophore in the UAF channel, the transformation and the PSF model for the supercritical channel, we then applied a second fitting step, in which we calculated the precise shape of the SAF PSF at the position of the fluorophore in the SAF channel from the SAF PSF model and use it to fit intensity and background fluorescence (assumed to be constant over the fitting ROI). This leads to a much more precise intensity estimation than during the first fitting step and allows for precise intensity estimation for fluorophores that due to their z position were too weak to be detected in the SAF channel in the first fitting step.

**Calculation of z positions.** From the relative intensities of the single fluorophores in the SAF and UAF channel we calculated $z_{dSALM}$ for every single-molecule blinking event according to Eq. (4). The astigmatism induced in the UAF channel helps in the initial fitting step by eliminating the degeneracy of the unmodified 2D PSF[17] and it extends the 3D resolution of dSALM beyond the range of SAF detection[18]. To combine $z_{dSALM}$ with the z position from the initial fitting of the experimental astigmatic PSF $z_{as}$, precise knowledge of the coverslip position is required. As this position changes due to drift, we first perform a 3D drift correction based on redundant cross-correlation[17]. Then, we plot for all localizations their intensity ratio $f^{ex} = N_S/N_U$ vs the drift-corrected $z_{as}^c$. Finally, we determine the glass position PSF $z_{as,0}$ from a fit of $f(z - z_{as,0})$ (Eq. 3) to these data points, treating $z_{as,0}$ as the only free fitting parameter. Using the respective CRLB as weights, we then combine $z_{dSALM}$ and $z_{as}^c$ as a weighted average to $z_{av}$:

$$z_{av} = \left(z_{dSALM} \cdot \delta z_{dSALM}^{-2} + z_{as} \cdot \delta z_{as}^{-2}\right) / \left(\delta z_{dSALM}^{-2} + \delta z_{as}^{-2}\right). \quad (13)$$

**Post-processing and rendering.** After fitting, localizations stemming from the same fluorophore and persisting over several frames were merged into a single localization. Localizations with a poor fit, resulting probably from overlapping localizations, were identified by their normalized log-likelihood value and filtered out (values lower than −2). In addition, dim localizations leading to a lateral

localization precision of less than 10 nm were filtered out. Superresolution images were reconstructed by rendering each localization as a Gaussian with a size proportional (factor 0.4) to the localization precision.

**Analysis of 30-nm DNA origami structures.** Structures were manually segmented. As the DNA origami structures can be tilted with respect to the coverslip, they were manually rotated to obtain side views using the 3D viewer in SMAP. Histograms of the rotated z position were fitted with a double Gaussian to extract the distance and the standard deviation of each ring.

**Analysis of 80-nm DNA origami structures.** We manually segmented 94 DNA origami nanopillars in 12 regions of interest. We then used the k-means clustering algorithm to assign the localizations to two clusters. We calculated the robust mean of the x, y, and z coordinates for each cluster. From the average positions, we calculated their distance in 3D and their projected distance in the x, y plane, and from these values the angle of the pillar with respect to the image plane.

**Reporting summary.** Further information on research design is available in the Nature Research Reporting Summary linked to this article.

## Data availability

Data are available from the corresponding author (J.R.) upon request. An example dataset including detailed instructions for analysis can be downloaded at https://www.embl.de/download/ries/dSALM/. Source data are provided with this paper.

## Code availability

The software for the data analysis used in this paper is available at https://github.com/jries/SMAP[39].

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

## Acknowledgements

We thank Maurice Kahnwald and Aline Tschanz for their help with sample preparation. We would like to thank Alexander Jesacher for his help with the CRLB and PSF calculations and Mathieu Ribes and Alexander Jügler for help with the production of the elliptical mirrors. This work was funded by the Deutsche Forschungs Gemeinschaft (DFG RI 2380/2), the European Research Council (CoG-724489 to J.R., StG-680241 to R.J.), the National Institutes of Health (U01 EB021223 to J.R.), the Human Frontier Science Program (RGY0065/2017 to J.R.), the Max Planck Society (R.J.), and the European Molecular Biology Laboratory (J.R., J.D., A.D., U.M.). S.S. acknowledges support by the QBM graduate school.

## Author contributions

J.R. and J.D. conceived the project. A.D., J.D. and J.R. built the microscope and performed and analyzed the measurements. U.M. prepared the samples. U.H., J.B. and R.H. produced the elliptical mirror. S.S. and R.J. provided the DNA-PAINT reagents. J.R., A.D., and J.D. wrote the manuscript with input from all authors.

## Funding

## Competing interests

The authors declare no competing interests.
