## [Peer Review File · Nature Communications]

Reviewers' Comments:

Reviewer #1:

Remarks to the Author:

Dear Editor and Authors,

Thank you for the opportunity to review this manuscript by Dasgupta and colleagues. The manuscript is well written, and the new methodological developments elegantly demonstrated. I am overall positive about its publication. I expect it to be particularly impactful in the bioimaging field, given the current emphasis on improving resolution in 3D fluorescence nanoscopy. That being said, I find the work somewhat too supported by simulations on the characterization side. On the other hand, the experimental work on real data is outstanding.

Below I provide some points for the authors to consider during revision:

Major #1: Since the paper is about true experimentally proven improvement in z resolution, I'd like to see Σ_z vs Z's experimental curves for the 4-fold range improvement claimed. To my best understanding, at $z < 250$ nm the authors use SALM, for $z > 250$ nm the authors use astigmatism, and then a mixture of the 2 measurements around $z \sim 250$ nm as seen in Fig. 3b. If so, please provide some further discussion to what happens in the transition regimes.

Minor #1: Please show the SAF PSF and how it varies experimentally with the z position.

Minor #2: Does the shown configuration only works with the NA1.7 objective due to the improved SAF channel signal? What would happen with NA1.49? It would be elegant if the authors could justify the use of the NA1.7 with experimental data showing an unacceptable level of noise and or diffraction signal from 1.49.

Minor #3: Why simulate a lower background noise in the SAF channel vs the UAF channel (10x less)?

Minor #4: It would be useful to have more information on where the EM was manufactured and code used (as mentioned in methods). Is the source-code available?

Minor #5: Please add a few comments about how sensitive the measurements are to the EM's alignment. Intuitively, I can easily imagine that the SALM PSF would dramatically be affected by alignment.

Minor #6: Figure 2 - can the authors comparatively show the astigmatism equivalent for the DNA origami rulers

Minor #7: Given the authors expertise in 4pi-SMLM, could you please add some text comparing the advantages and disadvantages of both methods against each other.

I hope these comments can help improve the manuscript.

With best regards,

-Ricardo Henriques

Reviewer #2:

Remarks to the Author:

“Direct Supercritical Angle Localization Microscopy for Nanometer 3D Superresolution”

In this manuscript, Dasgupta, et al. propose a refinement of supercritical angle localization microscopy (SALM), termed direct SALM or dSALM. In dSALM, supercritical angle fluorescence (SAF) and undercritical angle fluorescence (UAF) from fluorophores near a coverslip are directly split using an elliptical mirror (EM). By using an ultra-high NA objective and new fitting routines, the authors show that dSALM can achieve ~ 2.7 nm localization precision for ~ 5000 photons detected using Cramér-Rao bound (CRB) calculations. The authors then image DNA origami structures and clathrin coated vesicles and microtubules in cells, demonstrating the potential for dSALM for cell biology.

Overall, I believe dSALM is a significant improvement over previously demonstrated “virtual” SALM methods, e.g., SALM and DONALD, for the conceptual reasons stated by the authors. Further, the manuscript is clearly written and presented. However, I have significant concerns with the quantification and demonstration of dSALM, in that there is insufficient evidence to support the authors’ claimed resolution improvement and competitiveness with interferometric single-molecule localization microscopy (SMLM). dSALM performance, and any possible implementation difficulties, should be thoroughly reported so that the SMLM community can apply the technology effectively. I believe these concerns should be addressed in a revised manuscript before a final decision can be reached. Detailed recommendations for improvements are below.

Major comments

1. The authors note in several places (e.g., the abstract line 36 and introduction lines 90-92) that new fitting routines can extract precise intensities and, therefore, positions of single emitters. However, details of these routines are quite vague in the paper. I recommend the following:
 - a. Please add more details on the fitting procedures for determining intensity and background. How was the "precise shape of the SAF PSF" used to fit intensity and background? What assumptions were made regarding the shape and/or distribution of background fluorescence?
 - b. Please give examples of how the intensity and background of single-molecule localizations are computed, along with evidence that these estimates are accurate and precise. These estimates are critical for the accuracy of all SALM methods.
 - c. Please give mathematical details describing the "projective transformation" from the SAF to UAF channel.
 - d. Please give or refer to the exact equations used to compute axial positions z_{dSALM} (Eq. 4) and z_{av} (Eq. 13) in the main text.

2. Several claims are made regarding dSALM's experimental axial localization precision (2.7 nm, line 116) and resolution ("better than 10 nm," line 149). However, I generally find the evidence for these claims to be lacking.

- a. The localization precision of dSALM is never experimentally demonstrated, on DNA origami, fluorescent beads, or otherwise. Both Fig. 2e and Supp. Fig. 4 seem to report theoretical CRB values as experimental localization precision. This assumption is inappropriate, since CRB is the best-possible measurement variance of an unbiased estimator; no evidence was given to support a claim that the authors' fitting algorithms achieve CRB-limited precision.

The authors should either: 1) Show evidence that the fitting algorithms used in this paper can achieve CRB-limited precision in the context of dSALM, or,
2) Carefully characterize and report the performance of the proposed fitting algorithm using simulated dSALM images or analyses of dSALM localizations on DNA origami.

- b. More convincing evidence should be provided that dSALM can achieve <10 nm axial resolution, i.e., dSALM should be used to resolve features within experimental samples separated by less than 10 nm along the optical axis.
- c. I am concerned that the localizations reported by dSALM may be either unstable or inaccurate. Consider the DNA origami images in Fig. 2c. While there should be a clear hollow region in the middle of the origami when viewed from the top and side (the separation is ~30 nm), numerous particles show localizations within the structure. Compared to the finite localization precision (reported to be ~3 nm), the errors in localization (e.g., localizations in the middle of structures 2, 3, 7, and 8 when viewed from the top and in structures 1, 2, 3, and 6 when viewed from the side) are much larger. That is, there are variations in the details of these DNA origami that are not wholly captured by the localization precision (width of the Gaussian spots used for rendering the reconstruction).

Please rigorously quantify the accuracy of the proposed dSALM technique (hardware and software). Are there any biases in dSALM localizations? Where do these biases come from, e.g. sample tilt, SAF-UAF channel registration, etc., and how can they be mitigated? These considerations are critical to report for dSALM to be widely adopted by the SMLM community.

- d. Analysis of DNA origami localizations (inferred length d vs. tilt angle α , Fig. 2h) is used to claim that dSALM greatly reduces systematic errors compared to astigmatic 3D SMLM. However, no statistical analysis of the data was shown.
 - i. Since 3D localizations are available (from z_{aS} used in Eq. 13), the authors should compare z_{av} from dSALM to z_{aS} and explicitly show bias improvement comparing localizations of identical DNA origami particles.

- ii. Further, the authors should characterize more carefully the observed variations in origami distance, which seem to be much larger (~6 times, ~20 nm) than the stated localization precision (~3 nm). What variations arise from uncertainties in PSF calibration, sample tilt, etc.? What variations arise from the localization algorithm itself? What variations arise from the DNA particles?

Minor comments

1. Please state the number of photons needed to achieve the localization precisions quoted on line 112 of the main text.
2. As acknowledged by the authors, the NA 1.7 objective used in this work is highly specialized, requiring expensive coverslips, specific oil, and non-standard sample preparation and imaging protocols. To better put dSALM in context, please compute its theoretical precision for an NA 1.49 objective, which are much more common in the SMLM community. Compare the relative improvement of dSALM over vSALM for this NA value, and comment on what NA is required for dSALM to give a significant improvement over vSALM.
3. Please clarify the notation in Eq. (5), i.e., the matrix inverse should be computed first before the u^{th} diagonal entry is extracted:

$$\text{CRLB}_u = [\mathbf{F}\mathbf{I}^{-1}]_{u,u}$$

4. The use of “weighted average” on line 271 is confusing, since the actual weighted average equation has not been mentioned yet (Eq. 13). Please add a reference to Eq. (13) here.
5. Please clarify the meaning of “2 m” on line 310: Is this quantity the lens focal length or something else?
6. Please correct the following typos:
 - a. Fig. 2 caption “**d**, the histogram of...” should be “**e**, the histogram of...”
 - b. Check the consistency of the abbreviation of "liter" throughout the manuscript. Sometimes μL is used; sometimes μl is used, as on p. 11, line 353.

Reviewer #3:

Remarks to the Author:

It has been recognised for some time that comparing the undercritical angle fluorescence to the supercritical angle fluorescence provides a way of identifying the position of a fluorophore in z, because the supercritical angle fluorescence decays exponentially away from the coverslip. The approach has been pioneered by a number of labs, notably including the current authors and the Leveque-Fort lab. Here the authors present an improved approach to this idea, with two key aspects: splitting the light into the UAF and SAF components using an elliptical mirror and improved fitting. The first is important because the previous approach, which involved a more straightforward splitting of the light followed by blocking of the undesired component, discarded some photons.

The results are very impressive and overall I think the paper is an important contribution to the field and should be published in Nature Communications. However I do have some comments that I think should be addressed.

1) It is not made very clear what the limitations of this particular method are. Most of the relevant information is in supplementary figure 1, and there is almost no discussion in the text. There are a number of points that I think will be important for readers interested in the technique to understand which I don't think will be gained from the paper as currently written.

a) A large part of the resolution improvement here is coming from the combination of the very high NA objective and the DNA labelling method which allows collection of extremely high photon yields. Even when using only astigmatism a precision of around 15nm in z was expected. This is a long way above what can be achieved using most other labelling methods and probes. The authors highlight at the end that this method is not suitable for dSTORM, but theoretical calculations could still be carried out to show the likely performance using dSTORM and fluorescent proteins. (Given the current circumstances I am not going to suggest further experiments, though I think being able to see performance on more than one type of labelling would be helpful to readers and give them an idea of likely performance).

b) The authors highlight the issue with introducing astigmatism being the introduction of more uncertainty in either the x or y direction. However, since they have to introduce astigmatism they will suffer a similar disadvantage. Either this is not the case (in which case the paper is not clear) or Supplementary figure 1 is misleading because it gives the strong impression of entirely equivalent x and y precisions.

c) To emphasise the two points above: it is not made clear enough what elements of the high precision seen in the figures come from the labelling, which come from the optical setup and novel way of splitting the UAF/SAF light, and which comes from standard astigmatism with a very good setup and sample. It is really important for potential users of a technique like this for these different contributions to be clearly distinguished. It should be explicitly stated in the text that this method is only providing an advantage within the first ~300nm of the coverslip. This is not obvious from the paper as it stands.

2) I am not clear what is being presented in figure 2h. From the caption I had assumed that each point was an averaged set of data from a given sample ('an independent experiment') while the methods make it sound like each is one piece of DNA origami. If it's the former, I find the degree of scatter puzzling. If it's the latter, there aren't enough measurements to form a meaningful conclusion. If the performance of this method is so good, why is it only possible to get ~50 measurements? Are many images degraded (because of sample, optical or analysis issues) that the useful data is very limited? I would have expected a bare minimum of high hundreds of samples.

It is also important to note, when discussing the potential bias in z from using astigmatism in the

main paper, to note that the use of astigmatism with dSALM will introduce a similar problem.

Reviewer #1 (Remarks to the Author):

Dear Editor and Authors,

Thank you for the opportunity to review this manuscript by Dasgupta and colleagues. The manuscript is well written, and the new methodological developments elegantly demonstrated. I am overall positive about its publication. I expect it to be particularly impactful in the bioimaging field, given the current emphasis on improving resolution in 3D fluorescence nanoscopy. That being said, I find the work somewhat too supported by simulations on the characterization side. On the other hand, the experimental work on real data is outstanding.

We thank Ricardo Henriques for his constructive and useful comments.

Below I provide some points for the authors to consider during revision:

Major #1: Since the paper is about true experimentally proven improvement in z resolution, I'd like to see Sigma_z vs Z's experimental curves for the 4-fold range improvement claimed. To my best understanding, at $z < 250$ nm the authors use SALM, for $z > 250$ nm the authors use astigmatism, and then a mixture of the 2 measurements around $z \sim 250$ nm as seen in Fig. 3b. If so, please provide some further discussion to what happens in the transition regimes.

We now provide experimental data on beads quantifying the localization precision in z (see response 2 to reviewer 2 and Figure 2c). In addition, as suggested, we plot (for the experimental data) the z localization precision sigma vs z (Supplementary Figure 6), showing the fourfold resolution improvement, especially for the microtubule sample.

For thin samples (e.g. DNA origami structures) we only use the dSALM information. For extended data (clathrin, microtubules) we calculate, for the entire range, a weighted average of the SALM and astigmatism z-coordinates (eq. 13). As weights we use $1/\text{CRLB}$, our best guess for the variance of the z position. Thus, in the transition regime the z coordinates shift from the dSALM z values to the astigmatic z values. This is described in the methods section. We now add a direct cross reference on page 3 to the methods part and the equation.

Minor #1: Please show the SAF PSF and how it varies experimentally with the z position.

To measure the SAF PSF directly, we would have to move a very tiny but very bright bead through the sample, but we are not aware of such beads, and do not have means to position them so precisely. Instead, we now added a calculated side-view of the SAF PSF to Supplementary Figure 2, showing that the shape is quite constant with z, mostly the SAF intensity varies. The reason is, that due to the ring-mask the PSF gets Bessel beam like.

In addition, we now show an experimental PSF (UAF and SAF) obtained from a 100 nm bead by scanning the stage in z (Supplementary Figure 3). This is different to measuring the SAF PSF directly, but shows that slight defocus does not substantially change the shape of the SAF PSF.

Taken together, we assume during the fit that over the small range where we get a good SAF signal, the SAF PSF is mostly constant.

Minor #2: Does the shown configuration only works with the NA1.7 objective due to the improved SAF channel signal? What would happen with NA1.49? It would be elegant if the authors could justify the use of the NA1.7 with experimental data showing an unacceptable level of noise and or diffraction signal from 1.49.

In our opinion, it would be even more elegant if we could have shown that we can reach a similar resolution with the much more abundant NA 1.49 objective. A direct comparison is unfortunately not possible, as we do not have the elliptical mirror for the NA 1.49 objective. We now added CRLB calculations for the NA1.49 objective in the dSALM configuration to Figure 1 and Supplementary Figure 1. Compared to the NA 1.7 objective, the z resolution is a factor of 2-3 worse. What these

calculations however do not cover is that the much larger size of the NA 1.49 SAF PSF (compared to the NA 1.7 PSF) makes meaningful measurements complicated due to the much lower density of localizations and the requirement of a constant background over a larger area.

Minor #3: Why simulate a lower background noise in the SAF channel vs the UAF channel (10x less)?

Most cellular auto-fluorescence and all out-of-focus fluorescence cannot enter the SAF angles. Thus, the SAF channel should be almost background free. However, we still find some auto-fluorescence generated from the objective or optics.

Minor #4: It would be useful to have more information on where the EM was manufactured and code used (as mentioned in methods). Is the source-code available?

The source code is now added as Supplementary Material (CalcReflectivity.m). The mirror was produced by the co-author Uwe Hübner at the Leibniz institute. This information is now added to the methods.

Minor #5: Please add a few comments about how sensitive the measurements are to the EM's alignment. Intuitively, I can easily imagine that the SALM PSF would dramatically be affected by alignment.

The alignment is straight forward, as we image the BFP and can see both the BFP image as well as the mirror. Centering the mirror with respect to the BFP can thus be done with high accuracy. In addition, our mask is slightly larger than the critical angle, so even minor misalignments or changes in the refractive index (e.g. due to the presence of the cell) do not lead to undercritical light entering the SAF path. We added a comment about this to the Methods section.

Minor #6: Figure 2 - can the authors comparatively show the astigmatism equivalent for the DNA origami rulers

We want to stress that for Fig. 3 (previous Fig. 2) all z information is obtained only from the intensity ratio between SAF and UAF, no astigmatism z information was used. However, we still acquire the astigmatic images in the UAF channel. We now added an image comparing side views of the DNA origami based on astigmatism with dSALM (Supplementary Figure 7). As our astigmatism is rather weak, we cannot exclude that an optimized astigmatism could lead to a somewhat better z resolution.

Minor #7: Given the authors expertise in 4pi-SMLM, could you please add some text comparing the advantages and disadvantages of both methods against each other.

We now discuss the advantages and limitations of dSALM, also with respect to 4Pi-SMLM in the Discussion part.

I hope these comments can help improve the manuscript.

With best regards,
-Ricardo Henriques

Reviewer #2 (Remarks to the Author):

Review of Nature Communications manuscript NCOMMS-20-28873-T "Direct Supercritical Angle Localization Microscopy for Nanometer 3D Superresolution" In this manuscript, Dasgupta, et al. propose a refinement of supercritical angle localization microscopy (SALM), termed direct SALM or dSALM. In dSALM, supercritical angle fluorescence (SAF) and undercritical angle fluorescence (UAF) from fluorophores near a coverslip are directly split using an elliptical mirror (EM). By using an ultra-high NA objective and new fitting routines, the authors show that dSALM can achieve ~2.7 nm localization precision for ~5000 photons detected using Cramér-Rao bound (CRB) calculations. The authors then image DNA origami structures and clathrin coated vesicles and microtubules in cells, demonstrating the potential for dSALM for cell biology.

Overall, I believe dSALM is a significant improvement over previously demonstrated "virtual" SALM methods, e.g., SALM and DONALD, for the conceptual reasons stated by the authors. Further, the

manuscript is clearly written and presented. However, I have significant concerns with the quantification and demonstration of dSALM, in that there is insufficient evidence to support the authors' claimed resolution improvement and competitiveness with interferometric single-molecule localization microscopy (SMLM). dSALM performance, and any possible implementation difficulties, should be thoroughly reported so that the SMLM community can apply the technology effectively. I believe these concerns should be addressed in a revised manuscript before a final decision can be reached. Detailed recommendations for improvements are below.

We thank reviewer 2 for the comprehensive comments and hope that with additional experiments and clarifications we could resolve the concerns about the quantification.

Major comments

1. The authors note in several places (e.g., the abstract line 36 and introduction lines 90-92) that new fitting routines can extract precise intensities and, therefore, positions of single emitters. However, details of these routines are quite vague in the paper. I recommend the following:

a. Please add more details on the fitting procedures for determining intensity and background. How was the "precise shape of the SAF PSF" used to fit intensity and background? What assumptions were made regarding the shape and/or distribution of background fluorescence?

The spline fitting was developed by us in previous work (Li et al, Nature Methods 2018) and we refer to this in the methods section. We assume that the background is constant over the fitting ROI, but treat it as a free fitting parameter. We added this information to the text.

b. Please give examples of how the intensity and background of single-molecule localizations are computed, along with evidence that these estimates are accurate and precise. These estimates are critical for the accuracy of all SALM methods.

We now provide example data and detailed instructions on how to perform the analysis using our open-source software SMAP. We now quantify the localization precision experimentally using fluorescent beads (Fig. 2) and can show that this experimental localization precision is close to the theoretical localization precision calculated from the Cramer-Rao-Lower-Bound. This shows that the intensity and background values are accurate and precise, as otherwise the experimental localization precision would be far worse than the theoretical one.

c. Please give mathematical details describing the "projective transformation" from the SAF to UAF channel.

We now define the projective transformation in the methods.

d. Please give or refer to the exact equations used to compute axial positions z_{dSALM} (Eq. 4) and z_{av} (Eq. 13) in the main text.

We now refer to both equations in the main text.

2. Several claims are made regarding dSALM's experimental axial localization precision (2.7 nm, line 116) and resolution ("better than 10 nm," line 149). However, I generally find the evidence for these claims to be lacking.

a. The localization precision of dSALM is never experimentally demonstrated, on DNA origami, fluorescent beads, or otherwise.

We already quantified the experimental localization precision on the 30 nm DNA origamis by fitting the z-profile with a double Gaussian (new Fig. 3d, previously Fig. 2d). The width of the Gaussians is an upper limit for the experimental localization precision. We now add additional data in which we quantify the localization precision using fluorescent beads at different excitation intensities (Fig. 2).

Both Fig. 2e and Supp. Fig. 4 seem to report theoretical CRB values as experimental localization precision. This assumption is inappropriate, since CRB is the best-possible measurement variance of an unbiased estimator; no evidence was given to support a claim that the authors' fitting algorithms achieve CRB-limited precision.

The authors should either: 1) Show evidence that the fitting algorithms used in this paper

can achieve CRB-limited precision in the context of dSALM, or 2) Carefully characterize and report the performance of the proposed fitting algorithm using simulated dSALM images or analyses of dSALM localizations on DNA origami.

In Fig. 3d (previous Fig. 2d) and the figure legend we already characterize the performance on DNA origami and extracted experimental estimates for the localization precision. These can be compared with the calculated localization precision (Fig. 2e). We now added additional data taken on beads to directly compare the experimental with the theoretical precision (new Fig. 2) and show that the experimental localization precision gets very close to the CRLB-limited precision.

b. More convincing evidence should be provided that dSALM can achieve <10 nm axial resolution, i.e., dSALM should be used to resolve features within experimental samples separated by less than 10 nm along the optical axis.

Unfortunately, we are not aware of any commercial or easy to produce samples that have fluorophores spaced in z only 10 nm apart. DNA-origamis with such distances are all 2D samples. Thus we are afraid we cannot do the proposed experiment. But based on the theoretical and now also experimental characterization of the localization precision, 10 nm should be easily resolvable.

c. I am concerned that the localizations reported by dSALM may be either unstable or inaccurate. Consider the DNA origami images in Fig. 2c. While there should be a clear hollow region in the middle of the origami when viewed from the top and side (the separation is ~ 30 nm), numerous particles show localizations within the structure. Compared to the finite localization precision (reported to be ~ 3 nm), the errors in localization (e.g., localizations in the middle of structures 2, 3, 7, and 8 when viewed from the top and in structures 1, 2, 3, and 6 when viewed from the side) are much larger. That is, there are variations in the details of these DNA origami that are not wholly captured by the localization precision (width of the Gaussian spots used for rendering the reconstruction). Please rigorously quantify the accuracy of the proposed dSALM technique (hardware and software). Are there any biases in dSALM localizations? Where do these biases come from, e.g. sample tilt, SAF-UAF channel registration, etc., and how can they be mitigated? These considerations are critical to report for dSALM to be widely adopted by the SMLM community.

In Fig. 3d (previously 2d) we quantify the spread in z and find that it is on the order of the experimental localization precision (previous Fig. 2e) for the lower ring (used as a reference) and larger for the upper ring. This indicates deformations of the DNA-origamis. Tilt of the DNA origamis and possible deformations are additional contributors. Please note the large tail in localization precisions (Figure 2e, reaching 15 nm and above), easily explaining individual localizations that are offset from the structure. Please note that we render every localization as a Gaussian with a width $0.4 \times$ localization precision, as it has been shown (Baddeley, D., Cannell, M. B. & Soeller, C. Visualization of Localization Microscopy Data. MICROSCOPY AND MICROANALYSIS 16, 64–72 (2010)) that a rendering with a width equal to the localization precision deteriorates the resolution. This information is now added in the Methods sections. The registration is calculated based on the blinking fluorophores themselves and should not contribute much to any localization error. We cannot exclude that some localizations show a stronger error (e.g. because of activation of nearby fluorophores), but overall, the results in Fig. 2c (now 3c) are as expected. Indeed, compared to any 3D SMLM data found in the literature, this data shows a very high z resolution.

d. Analysis of DNA origami localizations (inferred length d vs. tilt angle α , Fig. 2h) is used to claim that dSALM greatly reduces systematic errors compared to astigmatic 3D SMLM. However, no statistical analysis of the data was shown.

We reported the Pearson Correlation Coefficient as a statistical measure for Fig. 2h (now 3h). We now added the distance measurement based on astigmatism, where we clearly find a systematic dependence of the length on the tilt angle.

i. Since 3D localizations are available (from z_{as} used in Eq. 13), the authors should compare z_{av} from dSALM to z_{as} and explicitly show bias improvement comparing localizations of identical DNA origami particles.

We now show for the 30 nm DNA origamis a direct comparison of z_{as} vs z_{dSALM} (Supplementary Figure 7). We find this more instructive than showing z_{av} , which is a weighted average of the z_{as} and z_{dSALM} .

ii. Further, the authors should characterize more carefully the observed variations in origami distance, which seem to be much larger (~6 times, ~20 nm) than the stated localization precision (~3 nm). What variations arise from uncertainties in PSF calibration, sample tilt, etc.? What variations arise from the localization algorithm itself? What variations arise from the DNA particles?

For the 30 nm origamis, the variation is comparable to the localization precision. This is indeed not true for the 80 nm samples. We note that these samples were very demanding to work with and it took a lot of effort to get the data in the paper (note that with additional experiments we could double the number of structures). Although they are commercially available, we could not buy them on the required high refractive index coverslips and had to prepare them ourselves. The yield was very low (typically only a handful of origamis per FoV), there were many origami fragments, and the origamis seemed to be not totally stable but to rotate around the attachment point over time. We thus attribute the larger error to a) this instability, b) a variability in the DNA origami, which are themselves not very rigid, in combination with the distribution of the binding sites, c) a fraction of localization events with close by binding events that lead to fitting errors, d) background binding events, e) for upright origamis, the z-resolution is already reduced. We think that with the additional experiments the data is still useful and demonstrates a reduction of systematic errors, but we could also remove the data or put it to the Supplementary section.

Minor comments

1. Please state the number of photons needed to achieve the localization precisions quoted on line 112 of the main text.

We added the number of photons per localization for Figure 2d (now 3d) to the figure legend.

2. As acknowledged by the authors, the NA 1.7 objective used in this work is highly specialized, requiring expensive coverslips, specific oil, and non-standard sample preparation and imaging protocols. To better put dSALM in context, please compute its theoretical precision for an NA 1.49 objective, which are much more common in the SMLM community. Compare the relative improvement of dSALM over vSALM for this NA value, and comment on what NA is required for dSALM to give a significant improvement over vSALM.

As suggested, we calculated the localization precision for dSALM using an NA 1.49 objective and added the graphs in Figure 1 and Supplementary Figure 1.

3. Please clarify the notation in Eq. (5), i.e., the matrix inverse should be computed first before the u th diagonal entry is extracted: $CRLB_u = [F^{-1}]_{u,u}$

We corrected the error

4. The use of “weighted average” on line 271 is confusing, since the actual weighted average equation has not been mentioned yet (Eq. 13). Please add a reference to Eq. (13) here.

We added the reference to eq. 13.

5. Please clarify the meaning of “2 m” on line 310: Is this quantity the lens focal length or something else?

We clarified the meaning.

6. Please correct the following typos:

a. Fig. 2 caption “d, the histogram of...” should be “e, the histogram of...”

b. Check the consistency of the abbreviation of “liter” throughout the manuscript. Sometimes μL is used; sometimes μl is used, as on p. 11, line 353.

We corrected the typos.

Reviewer #3 (Remarks to the Author):

It has been recognised for some time that comparing the undercritical angle fluorescence to the supercritical angle fluorescence provides a way of identifying the position of a fluorophore in z, because the supercritical angle fluorescence decays exponentially away from the coverslip. The approach has been pioneered by a number of labs, notably including the current authors and the Leveque-Fort lab. Here the authors present an improved approach to this idea, with two key aspects: splitting the light into the UAF and SAF components using an elliptical mirror and improved fitting. The first is important because the previous approach, which involved a more straightforward splitting of the light followed by blocking of the undesired component, discarded some photons.

The results are very impressive and overall I think the paper is an important contribution to the field and should be published in Nature Communications. However I do have some comments that I think should be addressed.

We also thank reviewer 3 for the encouraging and helpful comments.

1) It is not made very clear what the limitations of this particular method are. Most of the relevant information is in supplementary figure 1, and there is almost no discussion in the text. There are a number of points that I think will be important for readers interested in the technique to understand which I don't think will be gained from the paper as currently written.

We now discuss the advantages and limitations of dSALM, also with respect to 4Pi- SMLM in the Discussion part.

a) A large part of the resolution improvement here is coming from the combination of the very high NA objective and the DNA labelling method which allows collection of extremely high photon yields. Even when using only astigmatism a precision of around 15nm in z was expected. This is a long way above what can be achieved using most other labelling methods and probes. The authors highlight at the end that this method is not suitable for dSTORM, but theoretical calculations could still be carried out to show the likely performance using dSTORM and fluorescent proteins. (Given the current circumstances I am not going to suggest further experiments, though I think being able to see performance on more than one type of labelling would be helpful to readers and give them an idea of likely performance).

The calculations on Figure 1 are based on 5000 detected photons, which is what we typically detect under optimized dSTORM conditions. The improvement of dSALM vs astigmatism is mostly independent from the intensity, thus brighter or dimmer fluorophores would lead to a different scaling of Figure 1e, but the relative localization precisions among vSALM, dSALM and astigmatism would be very similar.

b) The authors highlight the issue with introducing astigmatism being the introduction of more uncertainty in either the x or y direction. However, since they have to introduce astigmatism they will suffer a similar disadvantage. Either this is not the case (in which case the paper is not clear) or Supplementary figure 1 is misleading because it gives the strong impression of entirely equivalent x and y precisions.

In principle, dSALM does not need astigmatism and the calculations based on theoretical PSFs in Supplementary Figure 1 do not assume astigmatism. In practice, we use weak astigmatism, also to extend the range of dSALM. This however is not necessary, and the astigmatism we use is weaker than typically used and leads to only a minor decrease in lateral resolution. We now updated Fig. 1 and Supplementary Fig. 1 to show different x,y localization precisions based on our experimentally calibrated PSF.

c) To emphasise the two points above: it is not made clear enough what elements of the high precision seen in the figures come from the labelling, which come from the optical setup and novel way of splitting the UAF/SAF light, and which comes from standard astigmatism with a very good setup and sample. It is really important for potential users of a technique like this for these different contributions to be clearly

distinguished. It should be explicitly stated in the text that this method is only providing an advantage within the first ~300nm of the coverslip. This is not obvious from the paper as it stands.

We now added (Fig. 1 and Supplementary Fig. 1) the calculations for dSALM with the 1.49 objective to directly illustrate the effect of a) different NA, b) vSALM vs dSALM and d) v/dSALM in comparison to astigmatism and the lateral resolution. We also modified the discussion to state clearly the limited range of dSALM.

Whereas the brightness of the fluorophores certainly helps in obtaining good experimental data (Figure 3, now 4), the relative improvement of dSALM with respect to astigmatism is largely independent of the intensity. The high NA and the new direct splitting contribute to a similar extent to the improvements.

2) I am not clear what is being presented in figure 2h. From the caption I had assumed that each point was an averaged set of data from a given sample ('an independent experiment') while the methods make it sound like each is one piece of DNA origami. If it's the former, I find the degree of scatter puzzling. If it's the latter, there aren't enough measurements to form a meaningful conclusion. If the performance of this method is so good, why is it only possible to get ~50 measurements? Are many images are degraded (because of sample, optical or analysis issues) that the useful data is very limited? I would have expected a bare minimum of high hundreds of samples.

It is also important to note, when discussing the potential bias in z from using astigmatism in the main paper, to note that the use of astigmatism with dSALM will introduce a similar problem.

We now reworked Fig. 2h (now 3h). As discussed in reply 2d ii to Reviewer 2, the 80 nm DNA origami structures were very difficult to work with, produced a low yield and quite some heterogeneity. Now we pool all different experiments and added additional data to show the results on 94 DNA origami structures, and show a direct comparison to the results achieved with astigmatism. As we typically get a low yield of about 5 structures per ROI and as the acquisition takes up to an hour, after which the immersion oil starts to degrade and has to be exchanged, it is not possible to increase the numbers much further or to filter for 'nice' structures. Thus, the limitations are mainly from the sample and not from the approach. We think that the new Fig. 3h demonstrates the point that dSALM is less affected by aberrations, but would be willing to remove the data altogether or put them to the Supplementary Data if the reviewer still finds them confusing or not of sufficient quality.

We now added a sentence that the astigmatic contributions are not free from systematic errors caused by depth-dependent aberrations.

Reviewers' Comments:

Reviewer #1:

Remarks to the Author:

Dear Authors and Editorial Board,

Thank you for taking our comments on board and extensively addressing the concerns raised. I believe the manuscript is now in good form for publication and would like to congratulate the authors for an excellent contribution to the imaging field.

“Direct Supercritical Angle Localization Microscopy for Nanometer 3D Superresolution”

In this manuscript, Dasgupta, et al. propose a refinement of supercritical angle localization microscopy (SALM), termed direct SALM or dSALM. In dSALM, supercritical angle fluorescence (SAF) and undercritical angle fluorescence (UAF) from fluorophores near a coverslip are directly split using an elliptical mirror (EM). By using an ultra-high NA objective and new fitting routines, the authors show that dSALM can achieve ~ 2.7 nm localization precision for ~ 5000 photons detected using Cramér-Rao bound (CRB) calculations. Measurements of DNA origami rulers show axial localization precisions of 3.2 nm and 5.8 nm using 21,300 photons, which is significantly worse than that predicted by CRB. The authors then image clathrin coated vesicles and microtubules in cells, demonstrating the potential for dSALM for cell biology.

Overall, this revision of the manuscript is significantly improved, especially in terms of simulation studies and imaging of fluorescent beads, and I believe dSALM has the potential to offer sizable advantages over previously demonstrated “virtual” SALM methods, e.g., SALM and DONALD. Further, the manuscript is clearly written and presented. However, I have significant concerns with the experimental quantification and demonstration of dSALM in biological samples, in that there is insufficient evidence to support the authors’ claimed resolution improvement. I believe this remaining concern should be addressed in a revised manuscript before a final decision can be reached. Detailed recommendations for improvements are below.

Major comments

1. Please provide data to support the statement in Line 142-143: “Compared to using Z_{dSALM} alone, the z resolution away from the coverslip is improved (Fig. 4b1).”
The authors should clearly illustrate which part of Fig. 4b1 is improved and quantify how much it is improved.
2. Please provide data to support the statements in Line 138-140 that the 150-nm spherical structure of clathrin coated pits and the 50-nm filaments can be resolved (Fig. 4).
How precisely does dSALM measure the size of these structures? If there is significant variance, then what are possible causes?
3. Please provide experimental evidence to support the statement in Line 164-165, “we developed dSALM, a 3D SMLM method that reaches a z resolution better than 10 nm near the coverslip.”
Demonstrating localization precision of less than 10 nm on 30-nm DNA origami rulers is not the same as sub 10-nm resolution.
Considering the difficulty in working with origami, I suggest that the authors can further quantify existing data, e.g. the 30-nm origami, to show the smallest structure in z direction can be resolved. For example, the “sides” of the upper and lower rings, which are separated by ~ 23 nm, in Fig. 3b should be easily resolvable by dSALM. What is the average and std. dev. of side lengths

measured by dSALM? If there are variations, what are the possible causes?

4. Please discuss variations in measured DNA origami lengths/separations in the main text.
 - a. There are excess localizations in the center and outside of the 30-nm origamis in Fig. 3c. Do these spurious localizations arise from structure variations? Overlapping blinking events? Difficulties in measuring the orientation of the structure itself? Problems with the fitting algorithm or channel registration?
 - b. Please state the average and std. dev. of the measured origami lengths d in the caption of Fig. 3h.
 - c. There is a large variation in the measured lengths d , beyond the stated localization precision, of the 80-nm origamis in Fig. 3h. Please discuss briefly in the text.
5. Please clarify the “more than 4-fold” resolution improvement over vSALM (line 168) along the z direction.
 - a. It is a bit difficult to interpret the right panel of Supplementary Fig. 6 (“resolution improvement”). It would be helpful for the authors to add an additional curve on top of the existing plot that represents the mean/median resolution improvement of dSALM over astigmatism as a function of z .
 - b. The best resolution improvement $\sigma_{\text{astig}}/\sigma_{\text{SALM}}$ occurs near the coverslip, as expected, and appears to be ~ 2 (panel a) and ~ 4 (panel b). Why is the improvement so different between clathrin and microtubule data? Does dSALM require a certain number of photons to achieve its best resolution improvement, and if so, why? For what percentage of localizations do you estimate the improvement to be “more than 4-fold?”
 - c. The comparison in Supplementary Fig. 6 is between dSALM and astigmatism alone, not dSALM vs. vSALM. I would expect vSALM to perform slightly better than astigmatism, thereby reducing the improvement of dSALM vs. vSALM. Please comment on exactly which data show dSALM’s four-fold improvement over vSALM.
6. In lines 232-233, the authors state, “Localizations are color-coded according to their z position zSALM,” but I cannot find the definition of these color codes. Please add colorbars to Figs. 3, 4, and S5.
7. Please clarify the “measured” data in Supplementary Fig. 3. The measured PSFs look almost too perfect: There is very little shot noise in each image, and the bead shown in each image is perfectly centered in x - y with respect to the camera pixels. Is it possible that these are not experimental images?

Minor comments

1. Please add scale bars to Fig. 1g.
2. Please add the texts of “side view” and “top view” to the figures in Fig. 4 and Supplementary Fig. 5, as in Fig. 3b,c,g.
3. Please add clarifications to the caption of Supplementary Fig. 2. For ‘SAF normalized’ x-z:
 - a. Is the mentioned “slice” oriented in the x-y plane?
 - b. Is the “summed intensity” taken across an entire x-y plane?
4. Please state how the intensities of PSFs in Supplementary Fig. 3 are normalized. Can I assume the three PSFs within one row ($z = 0$ nm, 250 nm, and 500 nm) are normalized individually? Is each x-y slice in the x-z images rendered with the correct relative intensity as a function of z?
5. The title of right panel in Supplementary Fig. 6a should be corrected to “resolution improvement.”

Reviewer #3:

Remarks to the Author:

I think the results are satisfactory and suitable to be published so the community can try the method for themselves.

The problems that the authors encountered with the samples emphasises the point that in many cases where you are aiming for this high a resolution the sample, not the imaging, becomes the limiting factor. Perhaps this is worth mentioning in the text?

In addition, I would note that it is useful to know about this regime, but it is also important for people to know the likely performance on more typical systems. The inclusion of the 1.49NA calculations is an improvement, but I still think that 5000 photons with optimal performance otherwise (i.e. negligible aberrations that degrade performance) is not representative of the majority of STORM experiments in biological samples.

Reviewer #3 (Remarks to the Author):

I think the results are satisfactory and suitable to be published so the community can try the method for themselves.

The problems that the authors encountered with the samples emphasises the point that in many cases where you are aiming for this high a resolution the sample, not the imaging, becomes the limiting factor. Perhaps this is worth mentioning in the text?

We agree that this point is worth mentioning and therefore added a sentence in the last paragraph of the main text.

In addition, I would note that it is useful to know about this regime, but it is also important for people to know the likely performance on more typical systems. The inclusion of the 1.49NA calculations is an improvement, but I still think that 5000 photons with optimal performance otherwise (i.e. negligible aberrations that degrade performance) is not representative of the majority of STORM experiments in biological samples.

In our experience, conventional dSTORM imaging with the dye Alexa Fluor 647 results in typically even more than 5000 photons per localization (after merging or grouping of localizations in consecutive frames belonging to the same fluorophore activation event). We recently quantified this systematically (Diekmann et al. Nature Methods 2020). Also, the famous Dempsey paper (Nature Methods 2011) lists a similar brightness. Certainly, the collected photons depend strongly on the microscope (filters, optics), the blinking buffer (thiol concentration, pH, BME vs MEA) and the fluorophore, but we believe that 5000 photons is reasonable and commonly achieved.

Reviewer #2

Review of Nature Communications manuscript NCOMMS-20-28873A “Direct Supercritical Angle Localization Microscopy for Nanometer 3D Superresolution”

In this manuscript, Dasgupta, et al. propose a refinement of supercritical angle localization microscopy (SALM), termed direct SALM or dSALM. In dSALM, supercritical angle fluorescence (SAF) and undercritical angle fluorescence (UAF) from fluorophores near a coverslip are directly split using an elliptical mirror (EM). By using an ultra-high NA objective and new fitting routines, the authors show that dSALM can achieve ~2.7 nm localization precision for ~5000 photons detected using Cramér-Rao bound (CRB) calculations. Measurements of DNA origami rulers show axial localization precisions of 3.2 nm and 5.8 nm using 21,300 photons, which is significantly worse than that predicted by CRB. The authors then image clathrin coated vesicles and microtubules in cells, demonstrating the potential for dSALM for cell biology.

Overall, this revision of the manuscript is significantly improved, especially in terms of simulation studies and imaging of fluorescent beads, and I believe dSALM has the potential to offer sizable advantages over previously demonstrated “virtual” SALM methods, e.g., SALM and DONALD. Further, the manuscript is clearly written and presented. However, I have significant concerns with the experimental quantification and demonstration of dSALM in biological samples, in that there is insufficient evidence to support the authors’ claimed resolution improvement. I believe this remaining concern should be addressed in a revised manuscript before a final decision can be reached. Detailed recommendations for improvements are below.

Major comments

1. Please provide data to support the statement in Line 142-143: “Compared to using zdSALM alone, the z resolution away from the coverslip is improved (Fig. 4b1).” The authors should clearly illustrate which part of Fig. 4b1 is improved and quantify how much it is improved.

We apologize that due to a reduced image quality in the converted PDF the difference is not clearly visible. We hope that this technical problem will be solved in the final publication. We now chose an example that shows an even larger difference and indicated the parts that show a noticeable difference with arrows.

2. Please provide data to support the statements in Line 138-140 that the 150-nm spherical structure of clathrin coated pits and the 50-nm filaments can be resolved (Fig. 4). How precisely does dSALM measure the size of these structures? If there is significant variance, then what are possible causes?

Clathrin-coated pits are known to continuously bend over time, with a wide distribution of sizes (Avinoam et al, Science, 2015). In figure 4 a-b, all visible structures are clathrin-coated pits or vesicles. The circularity is visible in the top view, the side view reconstructions clearly show circular and semi-circular structures with sizes in agreement with the range reported in the literature.

Likewise, our imaging approach can resolve the hollow structure of the immuno-labeled microtubule filaments as shown by the lower intensity in their center (side views). Since microtubules have a constant size, the slight differences between filaments can be due to tilts with respect to the plane used for reconstruction, as well as the randomness of labelling and the size of the antibodies.

Quantifying the sizes would not provide information about dSALM resolution or accuracy, but could be used to probe the structure and variability of coated pits and microtubules in the future.

3. Please provide experimental evidence to support the statement in Line 164-165, “we developed dSALM, a 3D SMLM method that reaches a z resolution better than 10 nm near the coverslip.” Demonstrating localization precision of less than 10 nm on 30-nm DNA origami rulers is not the same as sub 10-nm resolution. Considering the difficulty in working with origami, I suggest that the authors can further quantify existing data, e.g. the 30-nm origami, to show the smallest structure in z direction can be resolved. For example, the “sides” of the upper and lower rings, which are separated by ~23 nm, in Fig. 3b should be easily resolvable by dSALM. What is the average and std. dev. of side lengths measured by dSALM? If there are variations, what are the possible causes?

We agree with the reviewer that that localization precision and resolution are not the same and that it would be optimal to directly show the 10 nm resolution on a sample with 10 nm spacing in z. But we do not have access to such a sample. Still, we are convinced that we indeed reach this resolution based on our DNA origami and bead measurements. However, we now replaced the mention of resolution by “localization precision in z better than 4 nm”, something that we demonstrated throughout the manuscript.

Although slightly tilted, the 30-nm origami ruler sides are mostly influenced by the x-y resolution rather than the z resolution. Moreover, the lateral structure of the binding sites are not perfect squares and are helicoidal, causing additional variations. This prevents straightforward quantification, as opposed to the z dimension where a double Gaussian can be fitted. In our view estimating the distribution of sizes along z is the best quantification to experimentally determine our localization precision as it gives us an upper bound. The corresponding average and std deviation are mentioned already in the figure legend (Fig. 3).

4. Please discuss variations in measured DNA origami lengths/separations in the main text.
 - a. There are excess localizations in the center and outside of the 30-nm origamis in Fig. 3c. Do these spurious localizations arise from structure variations? Overlapping blinking events? Difficulties in measuring the orientation of the structure itself? Problems with the fitting algorithm or channel registration?

Spurious localizations are an unavoidable consequence of the stochasticity of SMLM, and can be found on every experimental image, but they are especially pronounced in DNA-PAINT. Here, imaging strands are known to unspecifically interact with DNA (see e.g. Geertsema et al,

<https://doi.org/10.1101/2020.03.28.010553>). In addition, we cannot rule out overlapping blinking events, although we tried to filter them out based on the loglikelihood value of the fit.

- b. Please state the average and std. dev. of the measured origami lengths d in the caption of Fig. 3h.

We added the average and std deviation for both astigmatism and dSALM.

- c. There is a large variation in the measured lengths d , beyond the stated localization precision, of the 80-nm origamis in Fig. 3h. Please discuss briefly in the text.

The 80-nm are not entirely rigid but can bend and have a certain degree of variability (Schmied et al, Nanoletters 2013). Then, the binding sites for the imaging strands have a certain distribution. Additionally, it is possible that imperfect fixation allowed small movements of the origamis. We added a sentence in the main text.

5. Please clarify the “more than 4-fold” resolution improvement over vSALM (line 168) along the z direction.

The more than 4-fold improvement is inferred from Supplementary Figure 1 (see answer to comment 5c).

- a. It is a bit difficult to interpret the right panel of Supplementary Fig. 6 (“resolution improvement”). It would be helpful for the authors to add an additional curve on top of the existing plot that represents the mean/median resolution improvement of dSALM over astigmatism as a function of z .

We now added the median of the localization precision/resolution improvement vs z as an overly.

- b. The best resolution improvement $\sigma_{\text{astig}}/\sigma_{\text{SALM}}$ occurs near the coverslip, as expected, and appears to be ~ 2 (panel a) and ~ 4 (panel b). Why is the improvement so different between clathrin and microtubule data? Does dSALM require a certain number of photons to achieve its best resolution improvement, and if so, why? For what percentage of localizations do you estimate the improvement to be “more than 4-fold?”

The median of the resolution improvement is around 4 for both data sets close to the coverslip. The localization precisions show a rather broad distribution. This could be a result of the random nature of photons per localization and of the background. We agree that the distribution of the localization precisions is different for both samples, and especially the astigmatism of the microtubule data set has a narrower distribution. This could be due to a slightly different position of the focus for astigmatism with respect to the coverslip, a different distribution of the target proteins and a different background in the cell. The resolution improvement should be independent of the brightness if we don't take into account the background, but in presence of background, it could depend on the signal to noise ratio.

- c. The comparison in Supplementary Fig. 6 is between dSALM and astigmatism alone, not dSALM vs. vSALM. I would expect vSALM to perform slightly better than astigmatism, thereby reducing the improvement of dSALM vs. vSALM. Please comment on exactly which data show dSALM's four-fold improvement over vSALM.

Indeed, the figure does not relate to the improvement of dSALM vs vSALM. However, our four-fold resolution improvement assessment does concern dSALM vs vSALM as it is based on Figure 1e and Supplementary Figure 1. In particular, Supplementary Figure 1e and 1f show the z localization precision ratios between the two approaches, yielding an average resolution improvement close to or better than 4-fold over the range 0-400 nm.

6. In lines 232-233, the authors state, “Localizations are color-coded according to their z position zSALM,” but I cannot find the definition of these color codes. Please add colorbars to Figs. 3, 4, and S5.

As suggested, we added the color bars.

7. Please clarify the “measured” data in Supplementary Fig. 3. The measured PSFs look almost too perfect: There is very little shot noise in each image, and the bead shown in each image is perfectly centered in x-y with respect to the camera pixels. Is it possible that these are not experimental images?

The measured PSF shown in Supplementary Fig. 3 is not an individual PSF, but an interpolated PSF computed from 40 bead stacks, as described in the Methods. Hence, the smoothness and alignment with the image pixels. In order to prevent any confusion, we changed the caption of the figure to emphasize this point.

Minor comments

1. Please add scale bars to Fig. 1g.

As suggested, we added scale bars to Fig. 1g.

2. Please add the texts of “side view” and “top view” to the figures in Fig. 4 and Supplementary Fig. 5, as in Fig. 3b,c,g.

We updated Fig. 4 and placed the text in the figure. Supplementary Fig. 5 already had the text for the panels, but we also now added ‘top view’ to the overview image.

3. Please add clarifications to the caption of Supplementary Fig. 2. For ‘SAF normalized’ x-z:
 - a. Is the mentioned “slice” oriented in the x-y plane?
 - b. Is the “summed intensity” taken across an entire x-y plane?

We now clarify these points in the caption.

4. Please state how the intensities of PSFs in Supplementary Fig. 3 are normalized. Can I assume the three PSFs within one row ($z = 0$ nm, 250 nm, and 500 nm) are normalized individually? Is each x-y slice in the x-z images rendered with the correct relative intensity as a function of z ?

Each PSF slice is indeed normalized individually, by its maximum amplitude (calculated PSF) or brightest pixel (measured PSF). We added this information to the caption of the supplementary figure in question.

5. The title of right panel in Supplementary Fig. 6a should be corrected to “resolution improvement.”

We corrected this mistake.

Reviewers' Comments:

Reviewer #2:

Remarks to the Author:

I thank the authors for responding carefully to my comments. My concerns have been adequately addressed in this revision.

It is my observation that the technical achievement of dSALM is somewhat held back by the difficulty of working with, and to some extent the unavailability of, imaging targets that can validate sub-10 nm resolution in 3D. As localization precisions go below 10 nm for these advanced methods, it is increasingly important to rigorously verify imaging accuracy at this length scale. It is a shame that further quantitative analysis to this effect is not available.

Reviewer #3:
Remarks to the Author:

I think the results are satisfactory and suitable to be published so the community can try the method for themselves.

The problems that the authors encountered with the samples emphasises the point that in many cases where you are aiming for this high a resolution the sample, not the imaging, becomes the limiting factor. Perhaps this is worth mentioning in the text?

In addition, I would note that it is useful to know about this regime, but it is also important for people to know the likely performance on more typical systems. The inclusion of the 1.49NA calculations is an improvement, but I still think that 5000 photons with optimal performance otherwise (i.e. negligible aberrations that degrade performance) is not representative of the majority of STORM experiments in biological samples.